# Review article: An interdisciplinary framework to support knowledge co-creation for drought impact assessments

Silvia De Angeli[1,2,3], Lorenzo Villani[4,5], Giulio Castelli[4,6,7], Maria Rusca[8], Giorgio Boni[3], Elena Bresci[4], Luigi Piemontese[4]

[1]Laboratoire Interdisciplinaire des Environnements Continentaux (LIEC), UMR CNRS 7360, Université de Lorraine, Vandœuvre-lès-Nancy, France
[2]Université de Lorraine, LOTERR, F-57000 Metz, France
[3]Department of Civil, Chemical and Environmental Engineering, University of Genoa, Genoa, Italy
[4]Department of Agriculture, Food, Environment and Forestry (DAGRI), University of Florence, Florence, Italy
[5]Department of Water and Climate (HYDR), Vrije Universiteit Brussel, Brussel, Belgium
[6]UNESCO Chair in Hydropolitics, University of Geneva, Geneva, Switzerland
[7]Environmental Governance and Territorial Development Hub (GEDT), University of Geneva, Geneva, Switzerland
[8]Global Development Institute, The University of Manchester, Manchester, UK

*Correspondence to*: Silvia De Angeli (silvia.de-angeli@univ-lorraine.fr)

**Abstract.** Drought impacts are increasingly recognised as socially influenced processes instead of mere hydro-climatic events, often resulting in fundamentally uneven outcomes across different social groups. Yet, drought impact assessments continue to be primarily based on hydrological modelling, which generates three major scientific gaps. First, they rarely include social science perspectives, thus largely overlooking the socio-political processes underlying drought propagation and the uneven distribution of its impacts. Second, these assessments are often developed through top-down modelling approaches, excluding the knowledge and perspectives of those who directly experience the impacts of droughts. Third, there is little consideration of the politics of knowledge production and how it shapes the model's assumptions and outcomes. We argue that there is a need for a transdisciplinary framework for drought impact assessment that can produce more power-sensitive, inclusive, situated, and reflexive assessments. Drawing from a diverse body of literature on transdisciplinarity in sustainability science, integrated water resources management, socio-hydrology, science and technology studies, and critical water studies, we developed an interdisciplinary conceptual framework to guide knowledge co-creation in drought impact assessment. By critically analysing this literature to identify important aspects that could enrich drought impact assessments from a transdisciplinary perspective, we identified and characterised five key dimensions: 1) the setup of a collaborative space, 2) the framing of the co-modelling process, 3) a shared knowledge of drought, 4) the co-selection and co-development of models to understand drought impacts, and 5) awareness of power biases and knowledge imbalances. While some of these dimensions are common to any transdisciplinary process, others are more specific to drought. Together, they represent a conceptual

framework to guide future developments in the field. We discuss our framework's applicability, limitations and contributions to advance transdisciplinary approaches in future drought impact assessments.

## 1 Introduction

Droughts are becoming increasingly widespread and impactful, with serious consequences for health, agriculture, societies, and the environment globally (Vicente-Serrano et al., 2021; Wilhite et al., 2007). Drought occurrences and impacts are generally considered hydrological extreme phenomena and, thus, are conceptualised and modelled with a hydrological process approach (Mishra and Singh, 2010, 2011). However, the social experience of drought is very different from how it is represented in hydro-climatic models (Enenkel et al., 2020; Kchouk et al., 2022). In fact, droughts are increasingly understood

as complex socio-hydrological phenomena that affect societies across interdependent sectors and socio-economic groups (AghaKouchak et al., 2021; G. Ribeiro Neto et al., 2023; Mehta, 2007; Van Loon et al., 2016b). Merely considering the physical dimension of drought neglects the interplay between the hazard and the socio-economic processes that make certain socio-economic groups, geographical areas, and urban and rural spaces more vulnerable and exposed than others (Hewitt, 2019). Recent research in climate justice and political ecology has built on this line of inquiry to conceptualise disasters as

generated by the interplay of hydro-climatic and historical, socio-political, economic, and institutional dynamics (e.g., Collard et al., (2018); Kallis (2008)). Drawing on these ideas, drought-related impacts, which often escalate into disasters, have been conceptualised as a social construction of water scarcity. This perspective highlights how different social groups experience the impacts of drought unevenly due to varying levels of power and influence. It also emphasises the variable "room for manoeuvre" of different socio-economic groups in responding and adapting to drought events. Finally, this perspective draws

attention to the underlying political and economic drivers, such as development pathways and policy visions, that shape the vulnerability of different groups and their exposure to hazards (Mehta, 2005; Rusca et al., 2023; Savelli et al., 2022; Usón et al., 2017).

The multidimensional nature of droughts has been addressed by recent interdisciplinary socio-hydrological research on water-related challenges, which aims at capturing the interplay between natural and social aspects (Rusca and Di Baldassarre, 2019;

Wesselink et al., 2017). Yet, there is an increasing recognition of the need to include societal perspectives within socio-hydrological modelling, such as those of non-academic actors directly experiencing the impacts of drought, through transdisciplinary studies (Arheimer et al., 2024; Hadorn et al., 2008). Transdisciplinary research brings "values, knowledge, know-how, and expertise from non-academic sources" (Klein, 2010) to the knowledge-creation process. This entails fostering mutual learning processes between science and society, reflecting a commitment to a science that collaborates with society

rather than simply serving it (Seidl et al., 2013). Transdisciplinarity includes a variety of approaches to knowledge co-creation or co-production (Bennich et al., 2022; Brugnach and Özerol, 2019; Norström et al., 2020). In the field of integrated water resource management, knowledge co-creation is often addressed by referring to the concepts of collaborative modelling or co-modelling (Basco-Carrera et al., 2017). This concept involves the collaborative construction of models, which can be physical,

conceptual, or computational representations of a system, process, or phenomenon. Co-creation provides the collaborative framework for ideation and value creation, while co-modelling offers the tools and methods to visualise, test, and refine these ideas into actionable solutions, enhancing the effectiveness of co-creation.

Recently, the co-creation of knowledge has been applied to hydrological sciences (Roque et al., 2022), especially within the socio-hydrology research niche, albeit with a prominent focus on flood risk and a limited application to drought (Vanelli et al., 2022). Examples of studies including a strong participatory component with non-academic stakeholders embrace co-modelling of i) water infrastructure and ii) water use under scarcity conditions. The first body of literature includes studies aimed at evaluating the feasibility of water infrastructure by integrating knowledge and mediating the values and expectations of different stakeholders (Coletta et al., 2024; Gil-García et al., 2023; Masi et al., 2024). For example, Gil-García et al. (2023) include experts' knowledge and opinions to co-design scenarios within their socio-hydrological modelling framework to build scenario assumptions within alternative adaptation strategies and guide policymakers when considering the construction of a dam in an unregulated basin. The second set of studies uses a variety of methods to provide guidance mediating the use of water resources under scarcity conditions (Baker et al., 2015; Gwapedza et al., 2024; Ocampo-Melgar et al., 2022; Rojas et al., 2022). For example, Liguori et al. (2021) explore a combination of storytelling and scientific data to guide the development of different co-designed narratives to support the planning of drought adaptation scenarios. The co-creation of adaptation scenarios is also the focus of a co-modelling approach proposed by Mustafa et al. (2021) to improve adaptation to hydrological extremes in the Limpopo River Basin. Although these approaches attempt to address drought impact assessment from a transdisciplinary perspective, they often limit the co-creation process to a specific phase of the whole transdisciplinary process, usually the definition of adaptation scenarios or the choice of indicators or model parameters (Luetkemeier et al., 2021).

An ample margin of investigation remains, especially in understanding the propagation of large-scale droughts or water scarcity into local impacts for an informed design of sustainable development policies (Pande and Sivapalan, 2017), for which transdisciplinary approaches can be pivotal. A comprehensive knowledge co-creation approach in drought research would require further efforts towards integrating different knowledge domains and fully engaging all actors throughout the knowledge-creation process. Advancing co-created research for drought impact assessment encounters specific barriers related to the different meanings of drought, its context-specific nature, its difficult predictability, as well as the subtle and unclear nature of its indirect impacts (Grainger et al., 2021).

Our paper advances the field of socio-hydrology by developing an interdisciplinary conceptual framework to guide hydrological and socio-hydrological modellers and practitioners in the co-creation of drought impact assessments. In this paper, the term "drought impact assessment" is used to refer generically to studies and projects that not only evaluate the hazard dimension of drought but also assess its impacts and support the identification and planning of drought management or adaptation measures. Moreover, this paper approaches drought from a socio-hydrological perspective, offering a framework tailored for transdisciplinary studies and projects that view drought as a result of feedback between water systems and human activities.

Given the limited literature on transdisciplinary approaches specifically focused on drought, we review and integrate knowledge developed in other scientific fields and disciplines. This allows us to identify recurrent themes across these disciplines, which can be considered key dimensions of a co-creation process for assessing and adapting to drought impacts. These dimensions are not sequential and do not need to be addressed in a specific order. Moreover, they are highly interconnected, and decisions or actions related to one dimension may iteratively influence others.

This work aims to enhance the understanding of co-creation in drought impact assessment by (i) identifying key dimensions necessary for ensuring the co-creation of knowledge in drought impact assessment and (ii) analysing the barriers and challenges to implementing co-creation in the context of drought impact assessment.

The paper proceeds as follows. In Sect. 2, we explain the methodological approach we followed in selecting the five bodies of literature, conducting the review, and identifying the key dimensions that informed our framework. Then, in Sect. 3, we present the interdisciplinary conceptual framework for transdisciplinary drought impact assessment, discussing each of the five identified key dimensions in detail. Next, in Sect. 4, we examine our framework's applicability and contributions to advancing transdisciplinary drought research, followed by a discussion of the limitations of transdisciplinary approaches to drought in Sect. 5. Conclusions are reported in Sect. 6.

## 2 Methods

We performed a literature review to inform the development of a conceptual framework for knowledge co-creation in drought impact assessments. Given the limited literature on transdisciplinary approaches specifically focused on drought, we reviewed and integrated knowledge developed in other bodies of literature.

We started our literature with co-creation experiences within Socio-hydrology. Socio-hydrology focuses on the dynamic interactions between hydrological processes and human behaviour, highlighting how social systems influence water resource management and vice versa (Di Baldassarre et al., 2013, 2015; Sivapalan et al., 2012). Our framework approaches drought from a socio-hydrological perspective, which is critical for understanding the socio-economic impacts of drought. Due to the limited literature available in the socio-hydrology field, specifically related to drought, we expanded the search by including the Integrated Water Resources Management (IWRM) body of literature due to its strong thematic relevance to drought impact assessment. IWRM promotes a holistic approach to managing water resources, considering the interconnectedness of environmental, social, and economic factors (Rahaman and Varis, 2005; Savenije, 1995; Savenije and Van der Zaag, 2008). IWRM frameworks are particularly relevant for drought impact assessments as they encourage collaborative decision-making and stakeholder engagement, which are crucial for effective drought management. Additionally, we included Sustainability Science (Brandt et al., 2013; Lang et al., 2012), with a focus on social-ecological systems research (Angelstam et al., 2013; Hummel et al., 2017), which has been foundational for the development of transdisciplinary approaches. This field emphasises the integration of knowledge across disciplines and sectors, fostering collaborative research that addresses complex environmental challenges like drought through the inclusive participation of non-academic actors. We initially identified a set

of key papers that addressed transdisciplinarity in the fields of Socio-hydrology, IWRM, and Transdisciplinary Sustainability SÂcience, and we then expanded the review by applying a snowball approach (Naderifar et al., 2017), i.e., by using the reference list or the citations to these papers to identify additional papers to investigate. As a result, we identified a first set of recurrent themes across these disciplines, which can be considered as key dimensions to define a co-creation process for drought impact assessment.

By expanding the research, two additional bodies of literature emerged as crucial to understanding power dynamics and vulnerabilities within the assessment context, as well as addressing ethical considerations: Critical Water Studies and Science and Technology Studies (STS). For the purpose of drought assessments, two interrelated areas of Critical Water Studies are particularly relevant: Hydrosocial Studies and Critical Disaster Studies. Hydrosocial Studies critique mainstream hydrological science for overlooking the power relations that shape the co-production of water and society. They challenge assumptions that solutions to water problems lie solely within technoscientific and hydraulic approaches (Boelens et al., 2016; Linton and Budds, 2014). Similarly, Critical Disaster Studies (Burton et al., 1993; White, 1945) argue that vulnerability to natural hazards, impacts, and disasters is deeply rooted in pre-existing inequalities and asymmetrical power relations. This field focuses on the political economy of vulnerability, examining how social and economic structures exacerbate uneven exposure and outcomes to natural hazards. It emphasises that "there is no such thing as natural disasters" (Smith, 2006). Together, these perspectives offer a historical and power-sensitive lens to examine water distribution, access, and water-related disasters. They underscore how water scarcity, droughts, and other water disasters are not merely natural phenomena but the outcome of deeply embedded unequal power dynamics and social vulnerabilities.

STS provides an insightful self-reflection of the co-production process. This body of literature challenges the notion of science as unbiased, highlighting that the process of knowledge development is influenced by power relations that determine which knowledge claims are considered valid and actionable (Budds, 2009; Goldman et al., 2019; King and Tadaki, 2018; Turner, 2011; Zwarteveen et al., 2017). This perspective emphasises the need for critical reflection on research practices, particularly regarding the inclusivity and legitimacy of knowledge in drought assessments. These two additional bodies of literature helped in expanding the previously identified dimensions. Moreover, they supported the identification of two additional dimensions, thus enriching the hydrological perspective with a more interdisciplinary description of drought impacts.

**3 Five key dimensions for drought impact knowledge co-creation**

The literature review led to the identification of five key dimensions for co-creating knowledge in drought impact assessment. These dimensions form a conceptual framework that provides the backbone of a knowledge co-creation process in the context of drought impact assessments.

**Table 1. Relationship between the literature in the five analysed bodies of literature and the dimensions of the framework they inform. Some studies appear in more than one body of literature due to their interdisciplinary nature.**

| Dimensions/ bodies of literature | Transdisciplinary Sustainability science | Socio-hydrology | Critical Water Studies (Hydrosocial Studies, and Critical Disaster Studies) | Science and Technology Studies (STS) | Integrated Water Resources Management (IWRM) |
|---|---|---|---|---|---|
| **Setting up a collaborative space for drought knowledge co-creation** (Sect. 3.1) | - Clarke and Clegg, 1998<br>- Clarkson, 1995<br>- Reed et al., 2009 | | - Rusca et al., 2023 | - Stirling, 2008 | - Basco-Carrera et al., 2017<br>- Hargrove and Heyman, 2020 |
| **Framing the drought co-creation process** (Sect. 3.2) | - Lillo-Ortega et al. 2019<br>- Mustafa et al., 2021<br>- Pham et al. 2020<br>- Sampson et al. 2020 | - Mustafa et al., 2021<br>- Pham et al. 2020<br>- Gwapedza et al. 2024 | - Ayantunde et al. 2015<br>- Thompson et al., 2017 | - Sarewitz and Pielke, 2007 | - Ballesteros-Olza et al., 2022<br>- Beek and Arriens, 2016<br>- Daré et al., 2018<br>- Nielsen-Gammon et al. 2020 |
| **Building a shared knowledge of drought** (Sect. 3.3) | - Grainger et al., 2021<br>- Gray et al., 2012 | - Rusca and Di Baldassarre, 2019<br>- Wesselink et al., 2017 | - Beck and Krueger 2016<br>- Kaika, 2003<br>- Mehta, 2001<br>- Rusca and Di Baldassarre, 2019<br>- Rusca et al., 2023<br>- Rusca et al., 2024<br>- Zwarteveen et al., 2017<br>- Wesselink et al., 2017 | - Alharahsheh and Pius 2020<br>- Beck and Krueger 2016<br>- Garb et al., 2008<br>- Krueger and Alba, 2022<br>- Landström et al., 2011<br>- Landström et al., 2023 | |
| **Co-selecting and co-developing models to understand drought impacts** (Sect. 3.4) | - Baumgärtner et al., 2008<br>- Biggs et al., 2021<br>- Iwaniec et al., 2020<br>- Raudsepp-Hearne et al., 2020<br>- Smetschka and Gaube, 2020<br>Hossain et al., 2020 | - Melsen et al., 2018<br>- Evers et al., 2012<br>- Evers et al., 2016<br>- Srinivasan et al., 2016<br>- Masi et al., 2024<br>- Piemontese et al., 2022<br>- Wens et al., 2020<br>- Fischer et al., 2021<br>- Baker et al., 2015 | | - Melsen et al., 2018 | - Basco-Carrera et al., 2017<br>- Carmona et al., 2013<br>- Kneier et al., 2023<br>- Singto et al., 2020 |
| **Accounting for knowledge biases and power imbalances** (Sect. 3.5) | | - Gwapedza et al. 2024 | - Alexandra and Rickards, 2023<br>Boelens et al., 2016<br>- Budds, 2009<br>- Collard et al., 2018<br>- Kallis, 2010<br>- Kallis, 2008<br>- Kaika, 2003<br>- King and Tadaki, 2018<br>- Krueger et al., 2016<br>- Macpherson et al., | - Goldman et al., 2019<br>Turner, 2011<br>- ter Horst et al., 2024 | - Basco-Carrera et al., 2017<br>- Falconi and Palmer, 2017 |

| | | | 2024<br>- Mehta, 2001<br>- Mukherjee, 2022<br>- Reed, 2008<br>- Rusca et al., 2024<br>- Savelli, 2023<br>- Sultana, 2020<br>- Swyngedouw, 2004<br>- Swyngedouw, 2009<br>- Thaler and Levin-Keitel, 2016<br>- Turnhout et al., 2020<br>- Zwarteveen et al., 2017 | | |
|---|---|---|---|---|---|

The framework is graphically depicted in Fig. 1 and outlines the five key dimensions of a transdisciplinary drought impact assessment. The five dimensions are the pillars of any transdisciplinary work,and include i) generic elements as well as drought-specific ones and ii) connection between the dimensions, showing how they mutually influence each other throughout a co-creation process. The dimensions are discussed in detail in Sects. 3.1 to 3.5, with explicit reference to the papers that informed each of them. Additionally, Table 1 connects the individual articles with the dimensions they contributed to and the bodies of literature to which they belong.

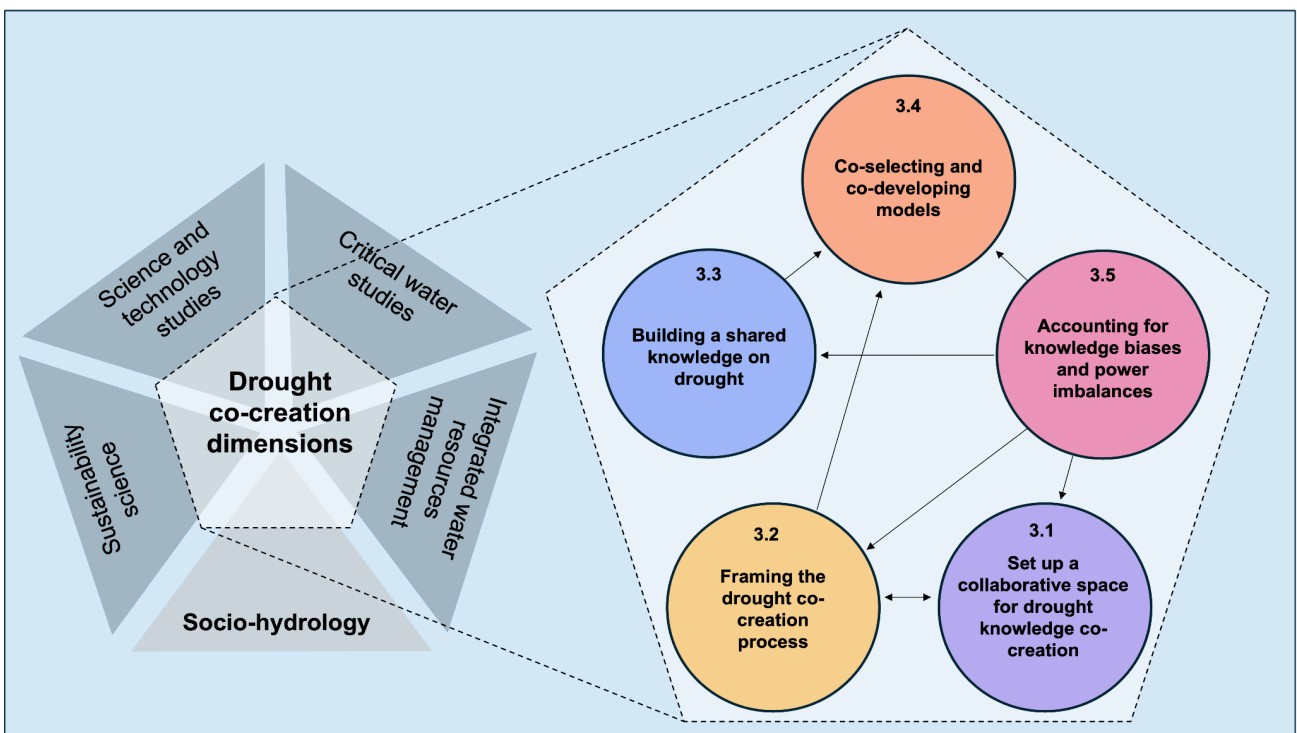

**Figure 1: Conceptual framework: A graphical representation of the theoretical background (on the left) and the five key dimensions of knowledge co-creation for socio-hydrological drought impact assessment, along with their interconnections (on the right).**

### 3.1 Setting up a collaborative space for drought knowledge co-creation

Setting up a collaborative space includes identifying, categorising, and establishing relationships among relevant stakeholders in the co-creation process (Reed et al., 2009). It ensures that all parties who are affected by drought or influence the mitigation process are involved in the decision-making and knowledge generation.

Stakeholders can be defined as "individuals, groups, and organisations who are affected by or can affect those parts of the phenomenon (this may include non-human and non-living entities and future generations)" (Reed et al., 2009). Stakeholders' identification is an iterative process where additional stakeholders are incorporated as the analysis unfolds. Setting clear boundaries for the study (Sect. 3.2) facilitates this process. Attention should be paid to verifying that these boundaries would not be too restricted to avoid the unintentional overlooking of some stakeholders, leading to the omission of relevant individuals associated with the phenomenon (Clarkson, 1995). Conversely, the boundaries cannot be too blurred. It is often impractical to include every stakeholder, requiring the establishment of well-founded criteria by the researchers and stakeholders to determine a cutoff point (Clarke and Clegg, 1998). A dynamic interplay between stakeholder identification and boundary-setting ensures that the co-creation process remains relevant and comprehensive without excluding important knowledge and perspectives.

Establishing a collaborative space is crucial for laying the foundations of a successful co-modelling process (Basco-Carrera et al., 2017). In transdisciplinary research, involving stakeholders serves multiple purposes (Stirling, 2008). Firstly, it upholds democratic ideals by emphasising inclusive processes. Secondly, it taps into stakeholders' insights and risk assessments to improve the quality of process outcomes. Lastly, it enhances the legitimacy of predetermined decisions, ultimately increasing their effectiveness in informing policy processes.

In addressing drought impacts, the failure to include marginalised groups—especially indigenous populations and low-income communities—can significantly hinder effective decision-making. Drought conditions often exacerbate existing inequalities, making it even more crucial to incorporate diverse perspectives and local knowledge into resource management strategies (Rusca et al., 2023). Despite claims of inclusive frameworks, available methods frequently fall short of genuinely integrating stakeholder input, leading to decisions that overlook local conditions and perpetuate power imbalances. A systematic approach to engaging all stakeholders, including those with indigenous knowledge, is essential for developing more equitable and effective responses to drought and ensuring that resource management is truly reflective of community needs and experiences (Hargrove and Heyman, 2020). Ensuring inclusivity is closely tied to reducing knowledge biases and power imbalances (Sect. 3.5). By actively involving diverse stakeholders, particularly those from historically marginalised groups, the co-creation process can challenge dominant narratives, integrate local knowledge and ensure that decision-making reflects a broader range of experiences, leading to more balanced and effective outcomes.

### 3.2 Framing the drought co-creation process

Framing the scope of the co-creation process involves establishing clear boundaries, both for the study itself and for the collaborative process (Daré et al., 2018). Additionally, it requires conducting a 'situation analysis' (Beek and Arriens, 2016),

which includes co-analysing the current situation, co-identifying key problems to address, and co-defining the main goals of the study.

The boundaries of the study define the specific aspects of the research that will be investigated. This includes the thematic boundaries, which determine the sectors, types of impacts, and affected units or groups that the study will focus on. For example, in a study of drought impacts, thematic boundaries might encompass sectors such as agriculture, urban water supply,

and public health, as well as the communities or populations that will be examined. Spatial boundaries define the geographical scale of the study, specifying the region, country, or ecosystem where the research would take place. Similarly, temporal boundaries specify the period covered by the research, whether it focuses on past events, current conditions, or future projections. In contrast, the boundaries of the co-creation process are concerned with how the research is conducted collaboratively. These boundaries define how the co-creation team will engage with the research, how they will contribute,

and how their input will influence the research outcomes. This includes setting the rules of engagement, determining how power and influence are distributed among stakeholders, and establishing decision-making processes to guide the collaboration.

Co-creation processes envision that not only methods, research itself, and interpretation of results, but also research questions are developed in partnership. From a transdisciplinary perspective, a crucial aspect of this step is aligning the research questions

with societal knowledge demands (Sarewitz and Pielke, 2007). Thus, framing the scope of the co-modelling process is crucial, as societal problems often lack clear boundaries, involve multiple stakeholders, and are deeply interconnected with other challenges, especially when dealing with complex and multifaceted phenomena, such as droughts. This step requires continual interaction and refinement, involving an iterative process of adding new stakeholders and adjusting the scope as the co-creation process evolves.

The iterative nature of co-creation, where stakeholders and research questions evolve over time, can lead to significant expansion, requiring considerable time, effort, and commitment from researchers, stakeholders, and funding bodies. This process can introduce delays and uncertainties in decision-making, particularly when addressing complex issues like droughts, where the intersection of environmental, social, and economic factors demands continuous reassessment. To keep the process manageable and focused, it is crucial to establish clear boundaries that define the limits of time, resources, and engagement,

ensuring that co-creation leads to meaningful and actionable outcomes (Thompson et al., 2017).

The framing of the boundaries of the study is particularly important in the context of drought impact assessment because the driving mechanisms of drought and its impacts, as well as drought governance strategies, can vary across spatial and temporal scales and sectors (i.e., health, agriculture, energy production, drinking water supply, etc.).

To set up the thematic boundaries, it is essential to co-defining the topics, themes, and areas of focus for the study. This

involves identifying the specific sectors, types of impacts, and the units or groups that will be affected by the study:

- **Sectors**: These are broad categories or fields that the study will address, such as agriculture, health, education, environment, drinking water supply, and energy production.

- **Types of impacts**: This includes the nature of the impacts the study aims to investigate or address, such as economic, social, or environmental impacts and the distribution thereof across space and socio-economic groups.
- **Impacted units or groups**: These are the specific entities or populations that will be affected by the study. They can be individuals, households, communities, organisations, or ecosystems.

Setting up the geographical boundaries requires the identification of the spatial scale in which drought impact assessment is performed, which can vary from local to global. The boundaries can be set considering physical (e.g., hydrological units or ecological systems – (Ballesteros-Olza et al., 2022; Mustafa et al., 2021)) or administrative boundaries (e.g., municipalities, countries, regions – Lillo-Ortega et al. (2019), Nielsen-Gammon et al. (2020)) as well as boundaries related to specific social-cultural-economic systems (e.g., Ayantunde et al. (2015), Pham et al. (2020)). Setting up the temporal boundaries refer to defining the specific period within which the drought impacts and mitigation strategy are studied. To illustrate, examining the impacts of a historical drought requires setting temporal boundaries to focus on a specific range of years or decades in the past. When studying the potential impacts of climate change on drought, temporal boundaries could be set to include projections for a future time horizon, such as the next 50 years (Sampson et al., 2020). Another relevant aspect of a transdisciplinary process for drought impact assessment is to perform a 'situation analysis' (Beek and Arriens, 2016), which encompasses (i) the co-analysis of the current situation (e.g. if drought is a current issue or a future concern, if mitigation measures already in place are effective and, if not, why, etc.), (ii) a co-identification of the main problems and issues to be addressed, as well as (iii) the co-definition of the main goals of the study. As an example, some co-modelling processes may aim at developing a shared vision of a water management plan (Gwapedza et al., 2024). Other studies might aim to predict future short-term impacts to suggest preventive strategies or quantify current and future drought impacts to define effective drought mitigation measures. Defining goals and outcomes together is essential to ensure transparency and prevent stakeholders' expectations from going unmet at the study's end.

The setting of the boundaries for the study itself and for the collaborative process are closely connected to two other dimensions of the co-creation process. To ensure that the research questions and scope align with societal knowledge demands, it's crucial to have a representative group of engaged stakeholders in the process (Sect 3.1). Framing the problem and setting the research agenda to encompass diverse understandings and perspectives may require expanding the team to include additional disciplines or engaging with other stakeholders to find the right mix. From a practical perspective, this would lead to an iterative initial phase in which a first set of stakeholders is identified, the scope of the process is framed, and then potential additional stakeholders can be added, requiring further refinement of the scope.

Moreover, the setting of the thematic, spatial, and temporal boundaries of the drought impact assessment study would finally influence the development of the co-modelling of the drought impacts (Sect. 3.4). These boundaries help define modelling scenarios that reflect diverse perspectives and align with the study's objectives. By establishing well-defined boundaries from the start, the co-modelling process can proceed smoothly and ensure the creation of contextually appropriate models.

## 3.3 Building a shared knowledge of drought

Successfully co-creating knowledge requires building a shared understanding of drought and its impacts (Grainger et al., 2021). This involves recognising that drought is conceptualised differently across disciplines, as well as between academia, practitioners and local communities. Thus, it is important to acknowledge that there is no single, universally accepted definition of drought (Krueger and Alba, 2022). Within hydrological sciences, which are often grounded in positivist paradigms (i.e., quantifiable through hydroclimatic thresholds), drought is defined as a geophysical phenomenon. In contrast, the interpretative and critical social sciences focus on the social construction of water insecurity and scarcity, examining the power relations and political economies that shape the uneven outcomes of droughts and the diverse experiences of their impacts (see, e.g., Mehta, 2001; Kaika, 2003; Alharahsheh and Pius, 2020; Rusca et al., 2023). These different conceptualisations are underpinned by distinct knowledge paradigms, which may hinder the development of inclusive and productive collaborations (Wesselink et al., 2017). Additionally, stakeholders who directly experience the impacts of drought, and who are involved in the co-modelling process, are likely to have alternative ways of knowing and defining drought, based on their "mental models" (Gray et al., 2012). For example, for urban dwellers in informal settlements, drought may be experienced and conceptualised as water shortages, water insecurity, waterborne diseases, or even as a source of physical and psychological stress, especially for women responsible for domestic water collection, rather than as a large-scale geophysical event (Rusca et al., 2023).

While the diversity of definitions and plural knowledges can complicate the process of co-creating a shared understanding of drought and its impacts (Landström et al., 2023), it also has the potential to generate a richer and more inclusive assessment. Thus, developing a shared understanding of drought should involve embracing and engaging with these different ways of knowing, rather than privileging one over another or creating a hierarchy between them. Moreover, as noted by Beck and Krueger (2016), depoliticised analyses of hydrological phenomena risk reproducing "authoritative representations of dominant perceptions of the world." Here, interdisciplinary assessments that work through epistemological and ontological differences have the potential to generate more nuanced and power-sensitive understandings of drought, exploring how power relations shape changes in hydrological flows and the distribution of hydrological risk (Rusca and Di Baldassarre, 2019), including drought risk. In this perspective, drought assessments can serve as boundary objects between different ways of knowing socio-climatic phenomena (Garb et al., 2008).

The process of developing a shared understanding of drought is closely intertwined with the co-modelling process (Sect. 3.4). On the one hand, co-modelling can serve as a tool to "redistribute expertise" (Landström et al., 2011), incorporating multiple perspectives and frameworks to create a more comprehensive and inclusive understanding of droughts. On the other hand, a shared knowledge of drought provides a crucial starting point for engaging in the co-modelling process. Thus, we view this as an iterative cycle in which building a shared understanding of drought (this Sect.) initiates the co-modelling process (Sect. 3.4), which, in turn, refines and deepens the shared understanding. This reiterative process also ensures that knowledge generation is dynamic and responsive to evolving perspectives.

Finally, the meaningful development of a shared definition of drought also requires addressing knowledge biases and power imbalances (Sect. 3.5). This involves confronting epistemic injustice by acknowledging that certain forms of knowledge on water, particularly those rooted in physical sciences, are often prioritised over other valuable knowledge systems, such as those held by local communities (Zwarteveen et al, 2017; Rusca et al., 2024). By recognising and addressing these imbalances, we aim to ensure that all forms of knowledge and actors are treated equitably in the co-creation of drought-related knowledge.

### 3.4 Co-selecting and co-developing models to understand drought impacts

Co-creating knowledge in drought impact assessments, and even more so in projected adaptation scenarios, very often relies on some levels of modelling (Baumgärtner et al., 2008). From a transdisciplinary perspective, co-modelling integrates non-scientist actors throughout the modelling process, irrespective of its purpose, whether forecasting, prescribing, explaining, describing, learning, or communicating (Srinivasan et al., 2016). With the term "model", we refer to any simplified representation of a real-life situation, thus including the whole range of qualitative conceptual models to predictive hydrological computational models. In transdisciplinary settings, the concept of "constructing models" can take on diverse interpretations. It can result in the co-creation of a conceptual model able to capture all the variables and processes relevant for describing the chain of problems and the study goals, or even include computations, algorithms, and dedicated modelling tools and platforms (Smetschka and Gaube, 2020). Whenever the co-modelling involves computations, algorithms, and dedicated modelling tools and platforms, stakeholders might be involved in the co-selection of the most appropriate tool, by considering not only the type of expected outcome but also the skills and background of the people involved in the modelling process, as well as other contextual factors related to the availability of economic resources and other implementation constraints. Even when consolidated models or software are preferred over fully co-created ones, setting-up and configuration tasks of the modelling process could include the participation of stakeholders to avoid modellers' pre-assumption and black box implementations (Melsen et al., 2018).

Through co-modelling, given a suitable interactive environment, non-specialized people can collaboratively produce models that are meaningful to them, fostering valuable discussions and the creation of new knowledge (Biggs et al., 2021). Flood risk management research demonstrates that collaborative modelling can improve planning, management, and social learning. In the DIANE project (Evers et al., 2012; 2016), this approach involves developing a shared understanding of flood risk, identifying objectives, and simulating alternative scenarios in workshops.

Drought impact co-modelling might also require a shared definition of modelling scenarios. Essentially, scenarios represent a collection of narratives or stories, which collectively depict various coherent future scenarios for a specific system (Biggs et al., 2021). A fundamental aspect of scenario development involves the co-creation of hypothetical future situations (Iwaniec et al., 2020; Raudsepp-Hearne et al., 2020), as well as conditions of the present or the past, that can be used for the co-modelling. These scenarios can encompass a range of variables, such as climate patterns, land use changes, socio-economic factors, or policy decisions.

In water scarcity and drought-related research, stakeholders ideally engage in participatory co-modelling by directly constructing models and tools, formulating scenarios and policy options, and assessing the effectiveness of identified solutions against jointly defined performance indicators or targets, often focusing on selecting alternative strategies or refining the application of technical solutions (Basco-Carrera et al., 2017). For example, Masi et al. (2024) applied an extended and highly participated multi-criteria decision-making analysis to optimize the siting of artificial reservoirs in Central Italy, while Piemontese et al. (2022) located best sites for sand dams in remote Angolan drylands with a similar approach. Wens et al. (2020) integrated an agent-based model with a process-based crop model to simulate alternative adaptation strategies, drawing from information collected through participatory methodologies. While these illustrative studies have the merit of including stakeholders' opinions, the conceptualization, modelling approaches, and sometimes even the proposed solutions, are established by the researchers - often hydrologists (Fischer et al., 2021). An interesting alternative was demonstrated by Baker et al. (2015) who applied a process-based hydrological model while involving stakeholders in the input generation phase. This approach emphasised the different needs and values of women and men in an Ethiopian catchment. Bayesian networks (e.g., Carmona et al., 2013; Kneier et al., 2023; Singto et al., 2020) and conceptual system dynamics models (e.g., Hossain et al., 2020) are often more suited to perform a transdisciplinary drought impact assessment because they facilitate the integration of knowledge from multiple disciplines and enable collaboration across sectors by modelling interdependencies. Moreover, Bayesian networks and system dynamics models account for uncertainty, feedback loops, and complex system interactions, making them suitable for implementing adaptive, context-sensitive adaptation strategies and responding to the multifaceted challenges of droughts.

## 3.5 Accounting for knowledge biases and power imbalances

This dimension focuses on the role of power and differential agency in shaping the co-creation of knowledge. We identify two key areas of influence. Knowledge biases refer to power dynamics within the knowledge production process itself. Mainstream interpretations of science often frame the development of knowledge as neutral, objective, and unbiased, especially in fields that prioritize quantitative methods. However, scholarship in science and technology studies (STS) and political ecology challenges this view, arguing that knowledge is shaped by power relations that determine which forms of knowledge and expertise are recognized as more scientifically valid and actionable (Budds, 2009; Goldman et al., 2019; King and Tadaki, 2018; Mukherjee, 2022; Turner, 2011; Zwarteveen et al., 2017; Rusca et al., 2024). Power imbalances refer to the unequal influence exerted by different stakeholders. Research in political ecology and critical disaster studies has shown how these power imbalances can generate and perpetuate uneven outcomes, including disparities in water allocations and adaptation to natural hazards across regions, intra-urban spaces, rural–urban populations, identities (e.g., gender, race) and income groups (see, e.g., Boelens et al., 2016; Collard et al., 2018; Sultana, 2020; Swyngedouw, 2004).

Overlooking knowledge biases and power imbalances risks marginalising certain actors and forms of knowledge while prioritising dominant, hegemonic ones in processes of knowledge co-creation (Krueger et al., 2016; Macpherson et al., 2024; Reed, 2008; Rusca et al., 2024; Thaler and Levin-Keitel, 2016). This is likely to undermine meaningful co-creation by silencing

alternative perspectives and reinforcing existing inequalities. Ensuring meaningful co-creation, therefore, requires reflecting on knowledge production itself, recognizing biases in what is deemed legitimate knowledge and acknowledging that co-creation *per se* does not inherently eliminate existing power dynamics. First, determining what counts as 'scientific' knowledge is particularly relevant in the context of co-creation for drought assessments. While co-modelling approaches have been argued to foster more inclusive and equitable knowledge, and thereby just water management (Basco-Carrera et al., 2017; Falconi and Palmer, 2017), it is the power-laden nature of scientific knowledge to avoid that scientific expertise dominates the co-creation process. Second, power imbalances among stakeholders can further undermine inclusive and equitable knowledge production processes. Dominant actors such as government agencies, international organisations, large NGOs, or scientific communities, typically possess more time and resources to lead participatory processes and set the parameters for participation. Their knowledge claims are typically regarded as more credible or valuable (Turnhout et al., 2020). Co-creation of water knowledge is also shaped by pre-existing relationships and histories of conflict (Budds, 2009), which are likely to influence the process. Embarking in a co-creation process requires engaging with different actors bringing their worldviews and goals. For instance, a drought can be framed differently, depending on the actors' histories, interests and experiences (Kaika, 2003). Powerful actors, like multinational, or political leaders, may exploit participatory context to legitimise hegemonic drought narratives that support specific outcomes or adaptation strategies, exacerbating the pre-existing power inequalities and benefitting those who are already more powerful (Alexandra and Rickards, 2023; Kallis, 2010, 2008; Mehta, 2001; Savelli, 2023). However, involving powerful stakeholders could be necessary to drive a change towards sustainable adaptation strategies. To that end, powerful stakeholders need to be made aware of the importance of engaging in the co-creation of solutions to drought. In fact, although benefiting from certain power dynamics in the short term, they could face long-term consequences as these dynamics evolve. For example, Gwapedza et al. (2024) show that, if large farmers are made aware of potential long-term drought impacts, they may remain interested in being engaged in the co-creation process and cooperating towards a shared drought adaptation plan.

This dimension acts as an ethical compass that fosters a power-sensitive and reflexive approach to the co-creation process. It is inherently relevant to all other dimensions of knowledge co-creation, as it seeks to dismantle the epistemic hegemony of scientific knowledge in drought assessments while incorporating diverse perspectives and ways of knowing throughout the co-creation process. First, recognizing power dynamics and epistemic biases is essential for establishing collaborative spaces (Sect. 3.1), which can only be truly inclusive if power imbalances are made explicit and addressed. Similarly, defining the scope of co-creation (Sect. 3.2) and building shared knowledge of drought (Sect. 3.3) often reflects the interests of powerful actors. A reflexive and power-sensitive approach seeks to pluralise narratives by creating space for alternative framings, especially those of marginalised groups. Last, a reflexive and power-sensitive approach is crucial for ensuring that co-modelling practices remain inclusive and equitable. As highlighted by ter Horst et al. (2024), this involves integrating reflexivity into every stage of the modelling process, from co-selecting and co-developing models to iteratively reviewing definitions, scenarios, and outcomes (Sect. 3.4).

## 4 How the framework can help advance drought research

This framework directly addresses three critical aspects, which are currently overlooked in hydrological approaches to drought impact assessment: the underrepresentation of socio-political factors, the exclusion of local knowledge, and the insufficient attention given to the politics of knowledge production.

Traditional drought assessments often fail to consider the socio-political contexts that influence vulnerability and impacts. To
bridge this gap, our framework emphasizes setting up a collaborative space for stakeholder engagement (Sect. 3.1), where all relevant actors, including marginalised groups, are involved in co-defining and assessing the drought impacts. This participatory and inclusive approach ensures that the socio-political factors, such as governance structures, power imbalances, and access to resources, are accounted for in the assessment process. Additionally, framing the scope of the co-creation process (Sect. 3.2) ensures that the research boundaries are collectively defined, incorporating socio-political factors that are crucial
for a comprehensive understanding of local drought impacts.

Moreover, many drought impact assessment studies overlook the knowledge and experiences of local communities, leading to assessments that are disconnected from the perception and expectations of those impacted. Our framework addresses this gap by suggesting a shared knowledge of drought (Sect. 3.3), which encourages the integration of local, indigenous, and community-based knowledge. The process fosters dialogue between different knowledge systems to create a more
comprehensive understanding of drought. In addition, co-selecting and co-developing models (Sect. 3.4) with stakeholders ensures that local knowledge is not only acknowledged but practically used in the development of drought impact models.

Lastly, drought impact assessments often neglect to analyse how power dynamics shape the production and use of knowledge. To address this, our framework recommends accounting for knowledge biases and power imbalances (Sect. 3.5), which critically examines how power relations influence both the co-creation process and the interpretation of outcomes. This
dimension encourages transparency about the political and social influences on knowledge production, ensuring that more powerful stakeholders, such as governments or large institutions, do not dominate the process. Furthermore, by fostering an inclusive, iterative process through framing the scope of the co-creation process (Sect. 3.2) and setting up a collaborative space (Sect. 3.1), the framework ensures that all voices, particularly those from marginalized communities, are heard and that knowledge production is not skewed by unequal power relations.

While the conceptual framework is valuable to everyone involved in assessing drought impacts, it is specifically designed to support 'positivist' hydrologists who may find it challenging to navigate between diverse scientific disciplines and within the extensive transdisciplinary literature. The proposed framework provides a foundational approach to help hydrologists effectively utilise transdisciplinary methods in the socio-hydrology of drought in a thorough and informed manner.

**Table 2. Core actions that define each of the five key dimensions of knowledge co-creation for socio-hydrological drought impact assessment and adaptation.**

| Dimension | Core actions |
|---|---|
| Set up a collaborative space for drought knowledge co-creation (Sect. 3.1) | • Implementation of a formal stakeholder analysis<br>• Definition of different levels of engagement for each stakeholder group<br>• Definition of clear strategies for stakeholder engagement |
| Framing the drought co-creation process (Sect. 3.2) | • Co-setting of study boundaries (thematic, geographic, temporal)<br>• Co-identification of key problems or criticalities<br>• Co-definition of study goals |
| Building a shared knowledge on drought (Sect. 3.3) | • Co-definition of 'drought' or 'water scarcity' concepts<br>• Co-definition of 'impact' concept<br>• Consideration and integration of different knowledge paradigms |
| Co-selecting and co-developing models to understand drought impacts (Sect. 3.4) | • Co-identification of the modelling processes<br>• Co-selection and/or co-development of tools and methods |
| Accounting for knowledge biases and power imbalances (Sect. 3.5) | • Explicit consideration of power dynamics among stakeholders<br>• Discussion and agreement upon modelling outcomes<br>• Measure in place to avoid power imbalances or biases among stakeholders |

The framework outlines crucial core actions across five key dimensions (Table 2), providing a foundation for translating these actions into practical protocols or guidelines for transdisciplinary studies focused on drought impact assessment and adaptation. While it provides a conceptual foundation, it is not a predefined operational protocol or a linear, step-by-step guide. Instead, it highlights essential considerations for protocol development, offering a flexible structure that can guide the creation of

440 tailored protocols. Furthermore, as illustrated in Fig. 1, the five key dimensions are deeply interconnected, requiring that any protocols developed based on this framework account for these interdependencies, potentially incorporating iterative cycles throughout the process.

Moreover, adopting the framework can support the advancement of impact-based drought forecasting by fostering standardised and scientifically sound collaboration with local communities. Incorporating indigenous knowledge addresses a current

limitation in understanding exposure, vulnerability, and local coping strategies. This integration is crucial for identifying

effective early actions and enhancing the overall response to drought impacts (Shyrokaya et al., 2024). Nevertheless, operationalizing this framework into a series of actionable steps and applying it to real-world case studies may present additional challenges due to the specificities of each context. Therefore, operationalization will require finding compromises to address context-specific constraints and limitations, which are discussed in Sect.5.

**5 Limitations of knowledge co-creation approaches**

In this work, we introduce a transdisciplinary approach to knowledge co-creation as a promising means to advance socio-hydrology research on drought. This approach aims to create knowledge that is not only useful and usable but also societally impactful. However, knowledge co-creation involves a series of inherent challenges and limitations that can significantly affect its application.

Including all relevant stakeholders or their representatives from a given "impacted group" — whether individuals, households, communities, organisations, or ecosystems (Sect. 3.2) — in the co-creation process is crucial. However, in cases where impacts are widespread over large spatial areas, the implementation of a fully representative co-creation process may be challenging due to the large number of people involved. This represents one of the limitations of applying transdisciplinary approaches. Furthermore, even when the number of participants is manageable, the process may still face obstacles due to a lack of economic resources, time constraints, or limited knowledge of the study area. Although highly motivated stakeholders are a prerogative to a successful co-creation approach, some stakeholder categories might not consider drought as an urgent problem or simply might not have enough time or interest in collaborating on the process. In some cases, the local political motivation and capabilities are key to ensuring a successful co-creation process. Vedeld (2022) explains the so-called "co-creation paradox", for which local political institutions that would benefit the most from co-creating solutions with local stakeholders, lack the political capacity and leadership to do so. Another key aspect of knowledge co-creation is acknowledging and involving holders of the different types of knowledge to ensure tackling the locally relevant problems and approaches (Brugnach and Ingram, 2012). Transdisciplinary studies require that goals and methodologies be collaboratively developed with stakeholders to ensure relevance, buy-in, and effectiveness. However, stakeholders are often only actively engaged after the research has received funding. By that point, the researchers have usually already defined key aspects, such as study goals and methodologies, which limits the scope for stakeholder input and collaboration. Donors' expectations might also limit the possibility of fully involving stakeholders in decision-making. Finally, the promoters of the co-creation approach - typically academics, institutions, and NGOs - have a specific background, which inevitably influences the whole process, either by prioritising some impacts, discipline, or sector as well as in the model selection, among other aspects (Melsen, 2022).

Co-creating knowledge has great potential to improve drought impact assessments and to promote proactive and effective drought management but cannot be considered a panacea. Some of the most relevant shortcomings related to co-creation are also related to the fact that it is considered an approach that is suitable for any context (Lemos et al., 2018). In particular, it is evident how the involvement of stakeholders in all kinds of research projects has become an indicator and a must-do by many

funding agencies, regardless of its suitability for a project (Cleaver, 1999; Spaapen and van Drooge, 2011). This, in turn, may lead to co-creation approaches that, due to time and resource limitations, tend to focus only on the same groups of stakeholders, prioritising familiarity over diversity, due to the high amount of time needed for building trust and engaged participants in the research approach (Porter and Dessai, 2017). This, in turn, can generate knowledge biases and exacerbate power imbalances (Sect. 3.5).

## 6 Conclusions

In this paper, we have argued that transdisciplinary approaches can significantly advance socio-hydrological drought impact assessments. Specifically, we proposed a drought-specific framework for knowledge co-creation, which offers a more comprehensive and inclusive method for understanding and modelling drought impacts. Traditional drought impact assessments typically focus on Earth system processes, often overlooking the critical socio-economic factors that shape vulnerability and exposure across different groups, regions, and communities. By incorporating these factors, transdisciplinary approaches can produce more holistic, power-sensitive, and reflexive assessments of drought impacts.

We have shown that transdisciplinary frameworks can advance socio-hydrological modelling by addressing key gaps in conventional approaches: the underrepresentation of socio-political factors, the exclusion of local knowledge, and the insufficient attention given to the politics of knowledge production. The framework we propose, structured around five key dimensions - (1) Establishing a collaborative space for stakeholder engagement, (2) Defining the boundaries and focus of the co-creation process, (3) Developing a shared understanding of drought, (4) Co-selecting and co-developing models to assess drought impacts, and (5) Addressing knowledge biases and power imbalances -, can guide scientists and practitioners in co-creating drought impact assessments that are more representative and contextually grounded.

The framework is particularly beneficial for 'positivist' hydrologists who might struggle with the broad transdisciplinary literature, offering a structured approach for using these methods in drought impact studies. It can represent a basis for the development of practical protocols and guidelines for transdisciplinary studies focused on drought impact assessment and adaptation. Furthermore, the framework aims to foster standard, scientifically sound collaboration with local communities, incorporating indigenous knowledge to improve understanding of exposure and vulnerability and enhance drought response strategies. This can ultimately improve impact-based drought forecasting.

However, co-creation is not without challenges, such as resource limitations, varying stakeholder motivation, and difficulties in achieving consensus. Though not a one-size-fits-all solution, knowledge co-creation, when thoughtfully applied, offers a promising pathway to proactive and effective drought impact assessment and adaptation.

*Data availability.* No data sets were used in this article.

*Author contributions.* S.D.A.: Conceptualization, Methodology, Investigation, Visualization, Writing – original draft preparation, L.V.: Conceptualization, Methodology, Writing – original draft preparation, G.C.: Methodology, Writing – original draft preparation, M.R.: Methodology, Writing – original draft preparation, G.B.: Writing – review & editing, Funding acquisition, E.B.: Writing – review & editing, Funding acquisition, L.P.: Conceptualization, Methodology, Investigation, Visualization, Writing – original draft preparation, Supervision.

*Financial support.* This study was carried out within (i) the RETURN Extended Partnership and received funding from the European Union Next-GenerationEU (National Recovery and Resilience Plan – NRRP, Mission 4, Component 2, Investment 1.3 – D.D. 1243 2/8/2022, PE0000005); (ii) the FSE REACT-EU, PON Ricerca e Innovazione 2014-2020 programme, co-financed by the European Union, Action IV.6 "Contratti di ricerca su tematiche Green" (CUP D31B21008660007).

*Competing interests.* The authors declare that they have no conflict of interest.

*Acknowledgements.* We thank Katja Malborg for sharing her experience and feedback on transdisciplinary approaches to knowledge co-creation during the early conceptual phase of the work.

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
