# Peer review of "Review article: An interdisciplinary framework to support knowledge co-creation for drought impact assessments"

_EGUsphere, 2024_

## Author Comment (AC1)

**REVIEWER 1**
* * *
**Comment 1**

I am aware that the manuscript was originally submitted as original research and might not have been intended as a review paper. While I agree with the editor's decision to consider it as a review, given that the methods clearly state they draw on existing literature, the manuscript still lacks substantial bibliographic evidence. The fields of literature referenced are vast, yet the specific studies informing the results are not identified, nor is it clear how these studies' and scientific fields' findings or philosophies contributed to the development of the proposed guidelines. This makes it difficult to see the manuscript as a well-founded review or a conceptual framework genuinely based on the literature.

**Response 1**

We understand the reviewer's concern about the unclear link between the cited literature and the five dimensions at the core of our framework. As the reviewer noticed, that is related to the original form of the submitted article, which was not meant to be a review paper.

**The papers that informed each of the five identified dimensions are directly cited in the paragraphs describing each dimension**, though this is not explicitly stated in the manuscript. We understand that this omission may create confusion for readers.

Responding more specifically to "how these studies' and scientific fields' findings or philosophies contributed to the development of the proposed guidelines",  we provide here an **explanation of our methodological approach**:

We started our literature with co-creation experiences within Socio-hydrology. Due to the limited literature available, specifically related to drought, we expanded the search by including the Integrated Water Resources Management (IWRM) body of literature due to its strong thematic relevance to drought impact assessment. Additionally, we included Transdisciplinary Sustainability science, which is foundational for the development of transdisciplinary approaches.
We initially identified a set of key papers that addressed transdisciplinarity in these three fields, and we then expanded the review by applying a snowball approach, i.e. by using the reference list or the citations to these papers to identify additional papers to investigate. As a result, we identified a first set of recurrent themes among these disciplines, which can be considered as key dimensions to define a co-creation process for drought impact assessment.

By expanding further the research, two additional bodies of literature emerged as crucial to understanding power dynamics and vulnerability within the assessment context, as well as addressing ethical considerations: Political Ecology**(*)** and Science and Technology Studies (STS). These two additional bodies of literature helped in expanding the previously identified dimensions. Furthermore, they supported the identification of two additional dimensions, thus enriching the initial more hydrologically-oriented perspective.

***(*)****Regarding that specific broad body of literature, we have specifically focused on Critical disaster studies (see answer to **Comment 9** for completeness).*

To improve the clarity of the reasoning and the connection between the reviewed literature and our results, we will **implement the following changes in the manuscript**:

1)  **We will include a "Methods" section**, which will describe our review approach and the reasons that brought us to draw from the mentioned bodies of literature.
2)  We will include a **table, which will link the individual article contributing to the conceptualization of each dimension to their respective body of literature** (among the 5 core bodies of literature in Figure 2). The table layout will appear as it follows:

| Dimensions/ bodies of literature | Transdisciplinary Sustainability science | Socio-hydrology | Critical disaster studies | Science and technology studies | Integrated water resources management |
|---|---|---|---|---|---|
| Set up a collaborative space for drought knowledge co-creation | | | | | |
| Framing the drought co-creation process | | | | | |
| Building a shared knowledge of drought | | | | | |
| Co-selecting and co-developing models to understand drought impacts | | | | | |
| Accounting for knowledge biases and power imbalances | | | | | |

3) We will explicitly **state** in the manuscript **that the different key papers which informed each dimension are directly cited in the paragraphs** describing the corresponding dimension:

*"Each dimension is discussed in detail in Sects. 3.1 to 3.5, with explicit reference to the papers that informed each of them. Additionally, Table XX connects the individual articles with the dimensions they contributed to and the bodies of literature to which they belong."*

**Comment 2**

Furthermore, there seems to be, at least to me, a strong disconnect between the research needs that justify this study and whether the proposed framework actually addresses these specific needs. The justification for developing a conceptual framework for transdisciplinary co-creation in drought impact assessment is because such frameworks are currently lacking. This lack arises from the complexities and challenges posed by both the subject and object of this study: transdisciplinary approaches (subject) and drought impact assessments (object). These complexities include, but are not limited to for transdisciplinary approaches, e.g. diverse stakeholder perspectives, power imbalances, communication barriers, project duration, and the need for building trust, among others. For drought impact assessment, these can be things such as high contextual variability, dynamism, multiple scales and levels of analysis, complex interactions between natural and social systems, and data availability, etc.—only to mention a very few.

And while I must point out that these limitations are indeed mentioned in the manuscript, it is only briefly, non-exhaustively, and not as central to the article, building on these gaps as they should be, or far too late, in Section 3.5 about transdisciplinarity, which appears more like a discussion than part of the results. In the absence of clearly identified research and practical gaps, and an explanation of how and why this study addresses them, the purpose announced is, in my view, defeated. Therefore, the proposed framework, in addition to not being sufficiently robust, lacks evidence that it could effectively address drought impact assessment.

**Response 2**

In this comment, the reviewer seems to criticize the unclear research gap that we're addressing, specifically the mismatch between our proposed framework compared to the mentioned gap.

In our revised version of the manuscript, after implementing the reviewers' comments, we will clarify that the **review aims at providing a theoretical guidance to approaching transdisciplinary drought co-creation research to socio-hydrologists.**

The justification for proposing such a framework is that droughts, unlike other natural hazards, are highly contextual, dynamic, and multi-scale, as the reviewer noted. This complexity necessitates a careful adaptation of traditional transdisciplinary approaches. Although drought complexity and subjectivity in impact assessment have been addressed in other research fields, in our framework we reconcile these two separate aspects, thus carving a specific framework to guide transdisciplinary research in drought impact assessment. In doing so, we specifically address the socio-hydrology niche, where we believe this framework can particularly be beneficial in advancing innovative approaches to drought research. In particular, we believe that the framework can be particularly beneficial to hydrologists, whose engagement in transdisciplinary research is still embryonic, but increasingly acknowledged.

We acknowledge that our intention does not read clearly from the introduction. To better clarify our research gap and justify how our framework will fulfill it we will do the following revisions:

1. In the abstract and introduction, we will **outline more explicitly the research needs** and the type of theoretical and methodological innovations/collaborations required to address these needs.
2. We will **merge section 2 "Knowledge co-creation in socio-hydrology of drought" to section 1 "Introduction"** to further contextualize the research gap within the target research field of socio-hydrology.
3. In the conclusion, we will **explicitly state how each dimension of the framework addresses specific limitations we identify in existing frameworks** (e.g., stakeholder participation addresses the exclusion of those most impacted and ensures a more comprehensive and context-specific understanding of the drought and its impacts; power-sensitive approach allows for greater awareness of power relations addresses the issue of limited engagement with the political and social aspects of droughts, whilst also providing essential inputs for ensuring effective and inclusive stakeholder participation).

Regarding point 1, here is the **proposed revision of the abstract,** which better aligns the research needs with the proposed framework, and reflects further changes that we will implement in the manuscript to address other specific comments from the reviewers (e.g., narrowing some areas of the literature and renaming certain key dimensions):

*"Drought impacts are increasingly recognised as socially influenced processes rather than mere hydro-climatic events, often resulting in fundamentally uneven outcomes across different social groups. Yet, drought assessments continue to be led primarily by hydrological scientists, which generates three major scientific gaps. First, they rarely include social science perspectives, thus largely overlooking the socio-political processes underlying drought propagation and the uneven distribution of its impacts. Second,*

*these assessments are often developed through top-down modelling approaches, excluding the knowledge and perspectives of those who directly experience the impacts of droughts. Third, there is little consideration of the politics of knowledge production and how it shapes model's assumptions and outcomes. We argue that there is a need for a transdisciplinary framework for drought assessment that can produce more power-sensitive, inclusive, situated, and reflexive assessments. Drawing from a diverse body of literature on transdisciplinarity in sustainability science, integrated water resources management, socio-hydrology, science and technology studies, and critical disaster studies, we develop a conceptual framework to guide knowledge co-creation in drought impact assessment. By critically analyzing this literature to identify important aspects that could enrich drought impact assessments from a transdisciplinary perspective, we identified and characterized five key dimensions: 1) the set up a collaborative space, 2) the framing of the co-modelling process, 3) a shared knowledge of drought, 4) the co-selection and co-development of methods to understand drought impacts, and 5) awareness of power biases and knowledge imbalances. While some of these dimensions are common to any transdisciplinary process, others are more specific to drought. Together, they represent a conceptual framework to guide future developments in the field. We discuss our framework's applicability, limitations, and contributions to advancing transdisciplinary approaches in future drought impact assessments."*

**Comment 3**

The suggested 'major dimensions' are quite general and could apply to any hazard or any aspect of natural resources management. They do not seem specific to drought impact assessment and are not innovative when compared to other frameworks for participatory modelling; for example, they could even fit as a subset of Ostrom's 8 rules for managing the commons.

**Response 3**

Our suggested dimensions do include some general aspects of transdisciplinary approaches since they were built on existing literature on transdisciplinarity. In fact, the purpose of our framework is to adapt generic transdisciplinary frameworks to the complex subject of drought impact assessment, which requires specific processes to be carefully addressed. As a consequence, some dimensions embrace those key features of any robust transdisciplinary process, including, for example, stakeholder engagement, while others are more drought-specific dimensions, for example, "building a shared knowledge on drought", which reflects the complex and context-specific nature of drought compared to other natural hazards.

However, we do believe that an effort to adapt an even more generic dimension to the specific subject of drought can indeed strengthen the relevance of our framework and increase its usefulness. For this reason, we will **revise the description of the different dimensions, highlighting the specific features connected to drought research**. We will also rename some of the dimensions to

explicitly include drought-related features. For example, "Mapping and engaging stakeholders" (Section 3.1) will be renamed as "Set up a collaborative space for drought knowledge co-creation." , "Framing the scope of the co-creation process" (Section 3.2) will become "Framing the drought co-creation process", while "Conceptualising and implementing the model" (Section 3.4) will be renamed as "Co-selecting and co-developing models to understand drought impacts".

Sections 3.1 and 3.2 address two dimensions that encompass key features of transdisciplinary processes. These sections would benefit most from revision to emphasize the specific aspects related to drought research. In Section 3.1 we will add the following text to better reframe this dimension from the perspective of drought impact assessment:

*"In addressing water resource challenges, particularly drought, the failure to include marginalized groups—especially indigenous populations and low-income communities—can significantly hinder effective decision-making. Drought conditions often exacerbate existing inequalities, making it even more crucial to incorporate diverse perspectives and traditional knowledge into resource management strategies. Despite claims of inclusive frameworks, available methods frequently fall short of genuinely integrating stakeholder input, leading to decisions that overlook local conditions and perpetuate power imbalances. A systematic approach to engaging all water stakeholders, including those with Indigenous knowledge, is essential for developing more equitable and effective responses to drought and ensuring that resource management is truly reflective of community needs and experiences (Hargrove and Heyman, 2020)."*

Moreover, we will start Section 3.2 providing more context about the importance of aligning the research questions with societal knowledge demands when addressing drought:

*"Co-creation processes envision that not only methods, research itself, and interpretation of results, but also research questions are developed in partnership. From a transdisciplinary perspective, a crucial aspect of this step is aligning the research questions with societal knowledge demands (Sarewitz and Pielke, 2007). Thus, framing the scope of the co-modelling process is crucial, as societal problems often lack clear boundaries, involve multiple stakeholders, and are deeply interconnected with other challenges, especially when dealing with complex and multifaceted phenomena, such as droughts. To ensure alignment between research questions and societal knowledge demands, it's crucial to have a representative group of engaged stakeholders (see Sect 3.1). Framing the problem and setting the research agenda to encompass diverse understandings and perspectives may require expanding the team to include additional disciplines or engaging with other stakeholders to find the right mix. From a practical perspective, this would lead to an iterative initial phase in which a first set of stakeholders is identified, the scope of the process is framed, and then potential additional stakeholders can be added, requiring further refinement of the scope."*

References:
    Hargrove WL, Heyman JM. A Comprehensive Process for Stakeholder Identification and Engagement in Addressing Wicked Water Resources Problems. Land. 2020; 9(4):119. https://doi.org/10.3390/land9040119
    Sarewitz, D., & Pielke, R. A. (2007). The neglected heart of science policy: Reconciling supply of and demand for science. Environmental Science & Policy, 10(1), 5–16. https://doi.org/10.1016/j.envsci.2006.10.001

**Comment 4**

This relates to another significant challenge for me, making the study quite difficult to follow: the lack of coherence and specificity in the terms, concepts, and definitions.

'Co-creation of knowledge': which knowledge related to drought?

'(Co)-modelling' is used too lightly and broadly: what type of models? What step? Models for what purpose or what rationale? Can we claim that all steps of all types of models for all purposes related to drought impact assessments should be co-created? If yes, why? This should be crystal clear early in the manuscript and not mentioned quickly as a form of discussion.

**Response 4**

Regarding the concept of 'co-creation of knowledge, in transdisciplinary research, knowledge emerges collaboratively through the engagement of researchers, practitioners, and stakeholders. This process ensures that diverse perspectives are considered, resulting in more relevant and practical solutions. Therefore, it would be contradictory to presuppose in advance the specific knowledge we aim to create.

Regarding the concept of 'co-modelling, as the reviewer points out, **we use co-modeling in a very broad way to accommodate different kinds of modeling approaches**. Here, a model is any simplified representation of reality, including mathematical or computer models, but also conceptual models related to hydrological or socio-hydrological aspects. We will reshape the dimension "Conceptualising and implementing the model" to **provide some practical examples of different models that can be applied to assess drought impacts**, as well as **detailing which specific modeling steps might benefit more from a transdisciplinary approac**h. We will particularly draw from case studies to showcase different ways of co-modeling within the socio-hydrological literature. Some examples of articles include the following:

Laura Gil-García, Héctor González-López, C. Dionisio Pérez-Blanco, "To dam or not to dam? Actionable socio-hydrology modeling to inform robust adaptation to water scarcity and water extremes", *Environmental Science & Policy*, Volume 144, 2023, Pages 74-87, ISSN 1462-9011, https://doi.org/10.1016/j.envsci.2023.03.012."

Virginia Rosa Coletta, Alessandro Pagano, Nici Zimmermann, Michael Davies, Adrian Butler, Umberto Fratino, Raffaele Giordano, Irene Pluchinotta, "Socio-hydrological modelling using participatory System Dynamics modelling for enhancing urban flood resilience through Blue-Green Infrastructure", *Journal of Hydrology*, Volume 636, 2024, 131248, ISSN 0022-1694, https://doi.org/10.1016/j.jhydrol.2024.131248.

Tracy J. Baker, Beth Cullen, Liza Debevec, Yenenesh Abebe, "A socio-hydrological approach for incorporating gender into biophysical models and implications for water resources research", *Applied Geography*, Volume 62, 2015, Pages 325-338, ISSN 0143-6228, https://doi.org/10.1016/j.apgeog.2015.05.008.

Matteo Masi, Chiara Arrighi, Francesco Piragino, Fabio Castelli, "Participatory multi-criteria decision making for optimal siting of multipurpose artificial reservoirs", *Journal of Environmental Management*, Volume 370, 2024, 122904, ISSN 0301-4797, https://doi.org/10.1016/j.jenvman.

**Comment 5**

And the proposed framework, which is quite broad, should also be more specifically tailored. The purpose of the co-creation process—'drought impact assessment and adaptation,' 'drought impact assessment and adaptation modelling,' 'support the planning of drought adaptation scenarios,' and 'drought assessment'—are used interchangeably.

**Response 5**

We agree with the reviewer that this multitude of different expressions might lead to confusion in the reader. The reason for that is because, with our framework, we aim at addressing all those studies and projects that not only evaluate the hazard dimension of drought but also assess its impacts and might even lead to support the identification and planning of drought management or adaptation measures.

We will **review the manuscript and make a more coherent use of these terms**. For example, we will replace the expression 'drought assessment' with 'drought impact assessment', or we will try to use the word adaptation without the word 'modelling' attached.

Moreover, in the introduction, we will explicitly state the following:

*"In this paper, the term "drought impact assessment" is used to refer generically to studies and projects that not only evaluate the hazard dimension of drought but also assess its impacts and support the identification and planning of drought management or adaptation measures."*

**Comment 6**

Similarly, 'major dimensions' and 'key dimensions' are used inconsistently. At certain points, vague terms or a list of very similar words make the text heavy and redundant (I will detail these in the specific comments), or terms from the social sciences jargon are used that are not suited for hydrologists, despite the study's aim to address both audiences.

**Response 6**

We thank the reviewer for pointing out this inconsistency. We will keep only 'key dimensions'. Additionally, we will thoroughly **revise the manuscript to ensure consistent vocabulary** and minimize the use of jargon.

**Comment 7**

Finally, the structure of the manuscript could be improved to enhance readability and robustness. The introduction is too long, lingers on concepts and explanations not pertinent to the comprehension of this study, and includes elements that would better fit in the sections on the state of the art or discussion. Conversely, the state of the art is incredibly short, lacks scientific evidence, and does not adequately guide the reader in understanding the concepts central to this study.

**Response 7**

We will substantially **reshape the introduction** by:

1) Reducing the unnecessary explanations on co-production, co-creation and co-modeling, including Figure 1 and Table 1.

2) Expanding the state of the art and research gap section, by a) incorporating part of the section "Knowledge co-creation in socio-hydrology of drought" and b) expanding the literature review on participatory approaches in socio-hydrological modeling.

**Comment 8**

There is no materials and methods section that provides the reader with the reasoning behind the development of the framework or the supporting evidence, which limits the study's replicability. Consequently, the results section is not appropriately supported. Something like Figure 2 should be in the methods section, not in the results.

**Response 8**

We will propose a **revised structure of the manuscript that includes a methods section** to thoroughly explain the review approach. Details about our methodological approach have been provided in **Response 1**.

We have also worked on a modified version of Figure 2 to address an additional comment (see our **response to Specific Comment SC26**). However, since the figure aims to illustrate not only the body of literature that informed the framework but also the five identified key dimensions, we believe it is more appropriate for it to remain in the section dedicated to the results, i.e. the presentation of the framework and its key dimensions.

**Comment 9**

Moreover, the sub-sections 3.1 to 3.5, which explain the major dimensions, contain elements related to defining concepts which should be in Section 2, or discussion elements that would be better suited for the discussion section. At the same time, the discussion could apply to any transdisciplinary approach rather than the specific conceptual framework proposed in the study, which should be a transdisciplinary framework for drought impact assessment. Also, the discussion contains claims, such as practical implications and benefits, which, in my opinion, are not substantiated by this study or are not evident. There is a notable underestimation of how significant the limitations are and how they affect the validity of the results. Some specific comments supporting these general observations are provided below. However, I suggest a thorough restructuring of the manuscript. This should include refining the research or practical gaps and objectives and narrowing down the research fields (e.g., 'science and technology studies,' 'political ecology,' etc., are too broad); or, if not narrowed down, it should be clearly explained what the contributions of these fields are to the framework.

**Response 9**

We thank the reviewer for this comment, which allowed us to better clarify which are the specific aspects in each of the selected bodies of literature that contributed to inform transdisciplinarity in drought impact assessment.

Socio-hydrology and Integrated Water Resources Management (IWRM) are those more thematically close to drought impact assessment. Socio-hydrology focuses on the dynamic interactions between hydrological processes and human behavior, highlighting how social systems influence water resource management and vice versa. Our framework approaches drought from a socio-hydrological perspective, therefore including this field is crucial for understanding the socio-economic impacts of drought. IWRM promotes a holistic approach to managing water resources, considering the interconnectedness of environmental, social, and

economic factors. IWRM frameworks are particularly relevant for drought impact assessments as they encourage collaborative decision-making and stakeholder engagement, which are crucial for effective drought management. Additionally, we included Transdisciplinary Sustainability science, which is foundational for the development of transdisciplinary approaches. This field emphasizes the integration of knowledge across disciplines and sectors, fostering collaborative research that addresses complex environmental challenges like drought through inclusive participation of non-academic actors.

Regarding political ecology, we agree it might be a quite broad field. Indeed, **our focus is mainly on Critical disaster studies.** Critical disaster studies is a well-established literature in environmental geography, political ecology, development studies and related fields that argues that vulnerability to natural hazards and related disasters is deeply rooted in pre-existing vulnerabilities, inequalities, and asymmetrical power relations. Critical disaster studies originate from the seminal work of Gilbert White and his colleagues in the 1940s. Their research redefined hazards as not just natural events, but as processes shaped by both environmental and social factors. Building on this foundation, scholars have since focused on the political economy of vulnerability, examining how social and economic structures exacerbate uneven exposure and outcomes to natural hazards and emphasising that "there is no such thing as natural disasters" (Smith, 2006). This perspective is key to understanding how uneven vulnerabilities are produced and maintained in the context of drought hazards, ensuring that the framework generates power-sensitive and inclusive drought assessments. A power-sensitive analysis works also as a tool to enhance stakeholder participation by shedding light on the underlying dynamics that may either hinder or facilitate meaningful engagement, ensuring that all voices are heard and considered in the assessment process

Eschewing mainstream views of science as unbiased and objective, Science and Technology Studies (STS) conceptualize the production and use of knowledge as shaped by power relations that determine whose knowledge claims should be deemed more relevant, scientific and actionable (Budds, 2009; Goldman, Nadasdy, & Turner, 2019; King & Tadaki, 2018; Turner, 2011; Zwarteveen et al., 2017). Recognizing that knowledge is always generated from specific perspectives and power-laden, it is crucial to critically reflect on the research process itself. This theoretical perspective encourages a reflection on research practices and on which forms of knowledge are valued, prioritized, and legitimized in the drought assessment process, ensuring that the assessment is both inclusive and aware of the power relations shaping its outcomes. Thus, **while critical disaster studies offer a framework to analyze power dynamics and vulnerability within the context of the assessment, Science and Technology Studies (STS) provides an 'ethical compass' for the research team.**

References:

White, G. F. Human Adjustment to Floods Department of Geography Research Paper No. 29 (Univ. of Chicago,1945).

Burton, I., Kates, R. & White, G. The Environment as Hazard (Oxford Univ. Press, 1993).

Smith, Neil. "There's no such thing as a natural disaster." Understanding Katrina: perspectives from the social sciences 11 (2006).

Budds, J., 2009. Contested H2O: Science, policy and politics in water resources management in Chile. Geoforum, Themed Issue: Gramscian Political Ecologies 40, 418–430. https://doi.org/10.1016/j.geoforum.2008.12.008

Goldman, M.J., Nadasdy, P., Turner, M.D., 2019. Knowing nature: conversations at the intersection of political ecology and science studies. University of Chicago Press.

King, L., Tadaki, M., 2018. A framework for understanding the politics of science (Core Tenet# 2). Palgrave Handb. Crit. Phys. Geogr. 67–88.

Turner, M.D., 2011. Production of environmental knowledge: Scientists, complex natures, and the question of agency. Knowing Nat. Conversat. Intersect. Polit. Ecol. Sci. Stud. Univ. Chic. Press Chic. 25–29.

Zwarteveen, M., Kemerink-Seyoum, J.S., Kooy, M., Evers, J., Guerrero, T.A., Batubara, B., Biza, A., Boakye-Ansah, A., Faber, S., Flamini, A.C., Cuadrado-Quesada, G., Fantini, E., Gupta, J., Hasan, S., Horst, R. ter, Jamali, H., Jaspers, F., Obani, P., Schwartz, K., Shubber, Z., Smit, H., Torio, P., Tutusaus, M., Wesselink, A., 2017. Engaging with the politics of water governance. WIREs Water 4, e1245. https://doi.org/10.1002/wat2.1245

**Comment 10**

The studies on which the framework was developed could be mentioned in the manuscript. For example, a list or table linking the studies, the field, and the specific results used for the framework could be provided in more detail as an annex and made available under the 'Data Availability' section.

**Response 10**

As mentioned in the response to **Comment 1**, we will include in the results section a table, which will link the individual article contributing to the conceptualization of each dimension to their respective body of literature (among the five core bodies of literature in Figure 2).
* * *
**Specific comments:**

**SC1** Title: The title can be rephrased to set the expectations of the content, with the mention that this article proposes a conceptual framework for co-creating knowledge. As it is now, I would expect this article to be about the outcomes of a co-constructed drought assessment. Furthermore, I do not see what the mention of 'socio-hydrology' in the title adds, nor why as it supposedly draws from four other fields.

**Response SC1**

We agree with the reviewer that the title might be improved to better reflect the aim of our work. Specifically, as it is currently phrased, it seems to present the outcomes of a co-constructed drought assessment rather than a conceptual framework for co-creating knowledge. For this reason, we propose rephrasing the title as follows:

**"Review article: An interdisciplinary framework to support knowledge co-creation for drought impact assessment"**

Regarding the possibility of removing the word "socio-hydrology" supported by the fact that it is only one of the five identified bodies of literature, we think that this suggestion might have been influenced by an unclear explanation of the role of socio-hydrology in our approach.

First of all, we would like to clarify that the reason for which we initially mentioned socio-hydrology in the title lies in the fact that **this paper approaches drought from a socio-hydrological perspective**, offering a framework tailored for transdisciplinary studies and projects that view drought as a result of feedback between water systems and human activities.

As a natural consequence, socio-hydrology is also one of the five investigated bodies of literature. Transdisciplinary approaches in socio-hydrology mainly address flood hazard. Nevertheless, as we intended to do with all the other bodies of literature, we tried to understand if some key features might be transferred to drought as well.

**SC2** L16: Transdisciplinary approaches to knowledge co-creation offer a promising opportunity to advance socio-hydrology

As developed later in the text, transdisciplinarity includes co-creation so this sentence is somewhat redundant or misleading.

**Response SC2**

Thanks for pointing this out. We will remove the expression "to knowledge co-creation", and rewrite the sentence as "Transdisciplinary approaches offer a promising opportunity to advance socio-hydrology".

**SC3** Furthermore, it is stated that the aim is to advance socio-hydrology, but then on L19, the contrast presented is about 'drought impact studies,' creating some confusion about the focus of the study.

**Response SC3**

We will rephrase as "advance drought impact studies in socio-hydrology".

**SC4** L32: Although originating from meteorological, hydrological or soil moisture anomalies, droughts are

This sentence is inaccurate. Please rephrase to correctly account for both the contributions of hydro-climatic and anthropogenic drivers to the occurrence of droughts.

**Response SC4**

We agree that this sentence is inaccurate. We propose to rephrase as follows:

*"Originating from the interaction of meteorological, hydrological or soil moisture anomalies with anthropogenic drivers, droughts are increasingly understood as complex socio-hydrological phenomena that affect societies across interdependent sectors and socio-economic groups."*

**SC5** L36: 'A rich geographical scholarship'

I suggest rephrasing as 'scholarship' is vague and many disciplines other than geography have 'theorised the interplay between hazards and the socio-economic processes that make some more vulnerable and exposed than others'. Alternatively, it could be deleted, as it does not add much.

**Response SC5**

We agree that this sentence is not adding so much to the previous one. Therefore we will remove the reference to the scholarship and rephrase as follows:

*"Merely considering the physical dimension of drought neglects the interplay between the hazard and the socio-economic processes that make certain socio-economic groups, geographical areas, and urban and rural spaces more vulnerable and exposed than others (Hewitt, 2019)."*

**SC6** L38 &40: I wouldn't use the word "climate justice and political ecology scholarship" and then give only two studies as an example.

**Response SC6**

We agree. We will remove the word scholarship from the sentence.

**SC7**

L39&41: Disasters or drought-related disasters, which are entire concepts on their own, are parachuted here, while previously, the text only focused on 'drought occurrences'.

**Response SC7**

We agree that the word "disaster" refers to a specific concept and can be misleading. According to UNDRR, a disaster is "a serious disruption of the functioning of a community or a society at any scale due to hazardous events interacting with conditions of exposure, vulnerability, and capacity, leading to one or more of the following: human, material, economic and environmental losses and impacts." Therefore, a negative impact cannot be automatically considered a disaster.

To be more precise, we propose to rephrase the manuscript as follows:

"*Drawing on these ideas, drought-related impacts, which often escalate into disasters, have been conceptualised as a social construction of water scarcity.*"

**SC8** L40-43: The sentence is convoluted and lacks clarity. It attempts to connect too many concepts—drought-related disasters, social constructions of water scarcity, differential agency, power asymmetries, uneven experiences of drought impacts, water (in)security, and political drivers—without a clear or coherent structure, making it difficult to understand the main point.

**Response SC8**

The original sentence was:

"*Drawing on these ideas, drought-related disasters have been conceptualised as a social construction of water scarcity to illuminate the differential agency and power asymmetries that determine highly uneven experiences of drought-related impacts and differential levels of water (in)security, and the underlying political drivers (Mehta, 2005; Rusca et al., 2023; Savelli et al., 2022; Usón et al., 2017).*"

We propose this version which improve the readability and understandability:

*"Drawing on these ideas, drought-related impacts, which often escalate into disasters, have been conceptualised as a social construction of water scarcity. This perspective highlights how different social groups experience the impacts of drought unevenly due to varying levels of power and influence. It also emphasizes the variable "room for maneuver" of different socio-economic groups in responding and adapting to drought events. Finally, this perspective draws attention to the underlying political and economic drivers, such as development pathways and policy visions, that shape vulnerability of different groups and their exposure to hazards (Mehta, 2005; Rusca et al., 2023, 2021; Savelli et al., 2022; Usón et al., 2017; Simon, 2014).*

**SC9** L47-56: The paragraph is very redundant. It repeatedly stresses the importance of interdisciplinary and transdisciplinary research without clearly distinguishing between the two or showing how they work together. The mention of the debate over the definition of transdisciplinary research seems unnecessary and does not directly support the main points about integrating perspectives and promoting mutual learning. Several parts convey overlapping ideas (e.g., "need for interdisciplinary research and participatory approaches" and "need to include societal perspectives"). To avoid redundancy, the paragraph could be condensed to focus more sharply on the main message: the importance of integrating various disciplinary perspectives and incorporating non-academic voices in drought research.

**Response SC9**

Based on the reviewer's comments, we have significantly shortened the section of the introduction dedicated to explaining the concept and aspects of transdisciplinarity. Additionally, we have better framed the need for interdisciplinary and transdisciplinary approaches to drought impact assessments to avoid overlapping or misleading sentences.

Hereinafter, we report the original paragraph with changes in track mode:

*"The multidimensional nature of droughts highlights the need for interdisciplinary  approaches to drought impact assessments. Interdisciplinary research on water-related challenges that cross natural and social sciences to capture this complex interplay is often invoked. Yet, studies that bring socio-hydrology into engagement with hydrosocial scholarship remain scant (Rusca and Di Baldassarre, 2019; Wesselink et al., 2017). Besides broadening the scientific knowledge domains, there is an increasing recognition of the need to include societal perspectives, such as those of non-academic actors directly experiencing the impacts of drought, within transdisciplinary studies (Arheimer et al., 2024; Hadorn et al., 2008). Transdisciplinary research brings  "values, knowledge, know-how and expertise from non-academic sources" (Klein, 2010) in the knowledge creation process. This entails fostering mutual learning processes between*

*science and society, reflecting a commitment to a science that collaborates with society rather than simply serving it (Seidl et al., 2013). Transdisciplinarity includes a variety of approaches to knowledge co-creation  (Bennich et al., 2022; Brugnach and Özerol, 2019; Norström et al., 2020)."*

**SC10**

L56-85 are not suitable for the introduction because they dive too deeply into some definitions, distinctions, and relationships between concepts, which can overwhelm the reader and diverge from the goal of progressing from a general overview of the state of the art to the identified research or practical gap. While it is important to provide enough detail about these concepts, this level of detail is more appropriate for a review on such concepts, or the methods or state-of-the-art sections.

**Response SC10**

We agree that this section is not suitable for inclusion in the Introduction. We will revise the Introduction accordingly by removing this detailed section and providing a more concise and focused summary.

**SC11** Considering that these paragraphs are kept and moved to more appropriate sections of the manuscript:

Figure 1, in my view, could be improved. While it conveys the idea of a concatenation of concepts, this is already explained in the text and does not need an illustration. The caption is somewhat repetitive:  the relationship is simply that they are concatenated. By the same logic, transdisciplinarity might be better represented as the largest encompassing square, also incorporating co-creation. The explanations for the 'co-' terms seem quite simplified, focusing more on what follows 'co-' than on the overall concept itself. Additionally, the figure is not referenced again in the text, raising questions about its necessity. In Table 1, more concepts are introduced—such as "co-design, co-dissemination, emergent and exploratory knowledge, traditional boundaries, co-implementing." If these terms are included (which I am not sure is necessary), it would be helpful to provide more detail on what they entail, and perhaps clarify the relationships between them and the ones in Fig1, which may not be solely horizontal.

**Response SC11**

We will remove Fig. 1. We agree that the concept conveyed by the figure regarding the concatenation of the terms is already addressed in the text.

**SC12** L80-83: I don't understand the mention of the scientific fields (sustainability, social-ecological systems, hydrological sciences, socio-hydrology) also considering that some (e.g. hydrology and SES) are not used for theoretical background.

**Response SC12**

We have mentioned social-ecological systems research as it is a specific niche within sustainability science, where transdisciplinary approaches are particularly popular. Similarly, we mentioned socio-hydrology as a niche of hydrology with ample examples of transdisciplinary literature. We will rephrase the sentence to make this link clear, as follows:

*"Traditionally, the fields of application of transdisciplinary research encompass sustainability science (Brandt et al., 2013; Lang et al., 2012), with a particular popularity within social-ecological systems research (Angelstam et al., 2013; Hummel et al., 2017). Recently, Co-creation of knowledge has also been applied to hydrological sciences (Roque et al., 2022), especially within the socio-hydrology research niche, albeit with a prominent focus on flood risk and a limited application to drought (Vanelli et al., 2022)"*

**SC13** L83: Finally, the research gaps or limitations for the development of a transdisciplinary framework for drought assessments are mentioned. In my view, the entire introduction should build towards these. However, these gaps or limitations are only briefly mentioned and not elaborated upon; they are a selection of many gaps and, most importantly, not specific to transdisciplinary approaches or drought assessments.

**Response SC13**

We have now reshaped the last part of the introduction by 1) reducing the space of the discussion on transdisciplinarity, 2) embedding the state of the art of socio-hydrology of droughts – including a broader mention to participatory and transdisciplinary approaches suggested by reviewer 2 – and 3) linking these two to the research gap.

We believe that these changes will make the introduction more focused and straight to the point.

**SC14** L86: Our paper advances the field of socio-hydrology by developing an interdisciplinary conceptual framework to guide scientists and practitioners in the co-creation of drought impact assessments.

Once again, why socio-hydrology? What are the boundaries of (i) socio-hydrology that (ii) this framework aims to expand or challenge? Additionally, it would be helpful to be more specific about what is meant by "practitioners", and I am uncertain whether conceptual frameworks that are not "practically" applied or ground-truthed can truly interest or benefit them. If so, please detail why and how.

**Response SC14**

We have substantially downsized the focus on socio-hydrology, starting from the title and including the removal of the section "Knowledge co-creation in socio-hydrology of drought".

As for the mention to practitioners, although the framework is rather conceptual, we believe it can serve as theoretical guidance to the development of practical protocols to be (ideally) carried out by researchers and practitioners together (as mentioned in L95).

**SC15** L87: "Given the limited literature on transdisciplinary approaches specifically focused on drought, we review and integrate knowledge developed in other scientific fields and disciplines."

The specific literature on transdisciplinary approaches focused specifically on drought is indeed limited. However, there are numerous studies, including reviews, on transdisciplinary frameworks for disaster management and integrated resources management. What I find puzzling is the decision to draw from other broad bodies of literature, rather than building on studies from the field of collaborative or participatory modelling, such as those related to floods (e.g., DIANE), integrated water management (WPI+), eutrophication (DEMO), and agricultural policy (SEAMLESS). This reasoning is not provided as there are no methods section.

In particular, WPI(+) is a highly developed, extensively studied, and widely applied inter- and transdisciplinary framework that addresses water scarcity and integrates all aspects mentioned in this study, including power asymmetries and the social construction of water scarcity.

**Response SC15**

We thank the reviewer for their thoughtful suggestions.

Regarding the SEAMLESS framework, we believe it may not be appropriate for inclusion in our review, as it primarily addresses governance and policy issues, which are outside the scope of our focus. On the other hand, the DIANE approach (Evers et al., 2012) offers valuable insights that would be relevant for the dimension related to co-modelling, as it specifically focuses on that aspect of co-creation.

The WPI is a well-established index that effectively captures the dynamics of water poverty. Although this framework can be helpful in providing an operational tool for water poverty assessment, we feel that it may not yet fully meet the criteria of a transdisciplinary framework or methodology, which is the focus of our review.

Unfortunately, we have been unable to locate the DEMO framework.

We would appreciate it if the reviewer provided us with more specific references and context for the mentioned frameworks.

**SC16** L89: The term "core aspects" in the introduction is quite vague and should be quickly defined or simplified here, rather than being left only to be exemplified in the results.

**Response SC16**

We agree that the term "core aspects" is quite vague. We will be more specific by rephrasing as:

*"This allows us to identify recurring themes across these disciplines, which can be considered key dimensions that can be effectively transferred and applied to define a co-creation process for assessing and adapting to drought impacts."*

**SC17** L90-95 are very confusing. The text states that the aim is to "enhance the understanding of co-creation" by identifying "key dimensions necessary for ensuring the co-creation of knowledge... through the development of an interdisciplinary framework." This creates a circular logic: the aim is to develop a framework, but then it says that developing the framework is part of the aim. The framework should be the outcome, not a part of the process for achieving the aim.

Also, the term "key dimensions" is used in this paragraph, whereas "major dimensions" was used earlier in the text. Please, be consistent with the terminology.

**Response SC17**

We agree that the text was somewhat confusing, as it did not clearly present the framework as a result. To improve clarity, we will rephrase it as follows:

*"This work aims to enhance the understanding of co-creation in drought impact assessment by: (i) identifying key dimensions necessary for ensuring the co-creation of knowledge in drought impact assessment (ii) analysing the barriers and challenges to implementing co-creation in the context of drought impact assessment. The identified key dimensions will represent an interdisciplinary conceptual framework, which will support scientists and practitioners in assessing the stage of advancement of co-produced drought impact assessment and directing the development of drought-specific co-creation protocols and tools."*

**SC18** L91: to what 'modelling' refers to?

**Response SC18**

We will rephrase the paragraph and remove the word "modelling". Please, refer to the **Response to SC17**.

**SC19** L95: 'development of drought-specific co-creation protocols and tool".

So this framework for knowledge co-creation would also guide practitioners to develop their own framework for knowledge co-creation? Additionally, what exactly are "co-creation tools"?

**Response SC19**

To clarify, the conceptual framework, which represents the core outcome of our literature review, is not an operational framework or protocol - i.e., it is not a list of sequential and mutually exclusive steps for conducting a transdisciplinary study. Instead, it condenses five key aspects that must be considered in developing protocols for transdisciplinary drought impact assessment. These aspects, referred to as "key dimensions," do not need to be addressed in a specific order. Additionally, they are highly interconnected, which means that protocols based on this framework must account for these interdependencies, potentially incorporating iterative processes throughout the steps.

To clarify and better set readers' expectations, we will include this explanation in the introduction.

**SC 20** Section 2 aims to present the state of the art but is quite short and misses crucial elements to fulfill this role. It consists of two paragraphs: the first highlights the slow shift in socio-hydrology research towards droughts and the need for more transdisciplinary studies. What constitutes the state of the art is the second paragraph, which only mentions three studies and focuses solely on recent research on collaborative modelling.

**Response SC20**

The original aim of Section 2 was to highlight how transdisciplinary approaches to knowledge co-creation in the socio-hydrology of drought are very limited, thereby justifying the need for our framework and the decision to expand the review to include other research bodies. However, in response to another comment from the reviewer (Comment 2), we will merge Section 2, "Knowledge Co-Creation in Socio-Hydrology of Drought," with Section 1, "Introduction," to better contextualize the research gap within the target field of socio-hydrology.

**SC21** L115-118: The statement, "Although these approaches advance the frontiers of co-modelling in drought research, they often allocate the co-design space to a specific phase of the drought knowledge generation process," contradicts the definition provided in Table 1. According to Table 1, co-modelling involves "decision-making processes in highly cooperative contexts (collaboration and/or joint action) with high levels of participation for key stakeholders in all phases of the modelling

process, including collaboration and joint action after the modelling process."

Also, what do "co-design space" and "drought knowledge generation process" mean?

**Response SC21**

We agree that this sentence was formulated in an unclear way and some terms were a bit obscure. We will rephrase the sentence as it follows:

*"Although these approaches attempt to address drought impact assessment from a transdisciplinary perspective, they often limit the co-creation process to a specific phase of the whole transdisciplinary process, usually the definition of adaptation scenarios or the choice of indicators or model parameters (Luetkemeier et al., 2021)."*

**SC22** L118: what is meant by "mature" in "A mature knowledge co-creation approach". Furthermore the second part of the this statement is assumed knowledge and lacks supporting evidence.

**Response SC22**

We will replace "mature" with "comprehensive." The need for comprehensive approaches is justified by the fact that current methods for drought impact assessment from a transdisciplinary perspective often limit the co-creation process to a specific phase of the overall transdisciplinary process (see **Response to SC21**).

**SC23** There is a clear absence of a section on materials and methods, or alternatively, methods and theory, that develop the theoretical framework underlying the conception of this study. While this is a critical review aiming to conceptually develop a framework, it does not mention any studies from which the framework key dimensions are identified, nor does it specify which aspects of the broad scientific fields the framework relies upon. There is no evidence that these dimensions are derived from an analysis of a comprehensive body of literature. Without such justification, the framework in Section 3 appears to be based solely on the authors' choices, which does not provide a robust foundation for a study, let alone a framework for practitioners.

I would expect for example a table with the following columns: Body of Literature/Scientific Field, Studies, Results of These Studies, and How These Results Lead to the Key Dimensions.

**Response SC23**

As mentioned in our Response to Comment 1, we will include a "Methods" section in the manuscript that describes our review approach and the rationale for drawing from the specified bodies of literature. Details about the review approach are

provided in the response to Comment 2, while a comprehensive explanation of our choice of specific literature is addressed in the response to Comment 9.

To enhance the clarity of our reasoning and the connection between the reviewed literature and our results, we will also include a table linking each article that contributed to the conceptualization of each dimension to its respective body of literature.

**SC24** L121: No evidence supporting this study being a critical review is provided.

**Response SC24**

We clarified that we conducted a critical literature review to specify that we did not perform a systematic review. By "critical literature review," we mean that we evaluated and synthesized the current state of knowledge from five different bodies of literature to address a specific need. To avoid any misunderstanding, we will remove it and just leave the generic term "literature review".

**SC25** L122: The term 'comprehensive analytical lens' is misleading, as drought risk, disaster, impacts, adaptation, and co-creation approaches are not examined, let alone in a comprehensive or holistic way. The complex issues related to these topics are not raised in the first place. This raises the question: what is the need for implementing dimensions 1-5? On what basis do these 'major dimensions,' presented as guidelines for implementing co-creation processes for drought assessment, originate? What is the justification for drawing from five broad fields of literature to derive these dimensions?

**Response SC25**

We agree with the reviewer that the term 'comprehensive analytical lens' might be misleading. We will rephrase suggesting that our framework provides the "backbone of a knowledge co-creation process" for hydrologists. Furthermore, we will review the manuscript and undertone some sentences to better align them with the aim of the work.

Moreover, as mentioned in our responses to previous comments, we plan to revise the manuscript to:

- Clarify the target audience of our work
- Clarify the research gaps and how our work will address them
- Provide a more explicit explanation of the methodology we followed
- Expand and strengthen the literature review that supports our findings
- Enrich the description of each dimension, including more examples of applications and ways to integrate each dimension into a transdisciplinary approach for drought impact assessment

We hope these revisions will help clarify many of the concerns raised in this comment, and we are confident that they will strengthen the manuscript overall.

**SC26** Figure2:

Now, what were 'major dimensions', then 'key dimensions', became 'drought co-creation dimensions'.

I don't understand the choice of using a Venn diagram. The Venn typically highlights a shared understanding or commonality between overlapping circles, which is not the case here. It is not my understanding that the commonalities between these bodies of literature lead to or support these dimensions. So I don't see the value in representing them this way.

**Response SC26**

We agree that representing the information with a Venn diagram might be misleading, as it implies that some bodies of literature overlap, which is not the case. We have therefore created the following updated version of the figure:

[Figure]

Please note that the final figure will be further improved. This is an initial version intended to better convey our idea.

**SC27** Why were these five broad scientific fields chosen? Why is the selection so general, and why couldn't the literature be more specific?

Beside the use of the words 'drought' and co-modelling (and as mentioned many times already, in the absence of strong evidence), I really don't see how this

framework is innovative or specific to drought impacts assessment.

Such guidelines can be found in the governance of the commons, in the steps for conceptual and computational modelling, or practically any transdisciplinary approach—by Google-ing the steps of a transdisciplinary approach. This is extensively covered in the literature. Is it truly revolutionary to suggest that, when conducting a (transdisciplinary) study, we identify the stakeholders involved, delineate its time and space boundaries, and collaboratively frame the problem? The real challenge is, in fact, how to implement these steps effectively, and this is also well-covered in the literature.

**Response SC27**

To clarify the points that the reviewer raises in this comment, we have added a methods section addressing the choice of the specific bodies of literature and the analytical procedure that we followed for the review. Briefly, we started from co-creation experiences within socio-hydrology, and we then expanded the search by including early transdisciplinary work in sustainability science and to water-related fields across different disciplines. For this reason our framework necessarily reviews and includes some key and common co-creation steps, such as stakeholder involvement, but it also extends beyond by including specific social sciences fields like critical disaster studies and science and technology studies, which altogether for a much more comprehensive set of interdisciplinary knowledge to guide knowledge co-creation in drought research.

Additionally, in response to other comments, we will adapt the more generalized elements of our framework to focus specifically on drought. We will revise the descriptions of the different dimensions to emphasize features unique to drought research, enhancing the relevance and specificity of our work.

**SC28** From L131 to 266, the five dimensions are described in very general terms with broad recommendations but lack practical or specific guidance for implementation in the context of drought. Additionally, there are numerous inaccuracies and contradictions between the underlying processes of drought and drought impacts and the key dimensions that are supposedly designed to address these issues. Moreover, there are several elements of general discussion that are misplaced and do not fit within this section.

**Response SC28**

We will revise the five dimensions to target more specifically drought impact assessments, while rephrasing them to be concise and avoid over generic recommendations.

**SC29** How does it make sense that accounting for power imbalances and knowledge biases is addressed last among these dimensions, while the very first dimension is stakeholder identification and mapping? For example, L137 mentions: "[involving stakeholders] … upholds democratic ideals by emphasising inclusive processes. Secondly, it taps into stakeholders' insights and risk assessments to improve the quality of process outcomes. Lastly, it enhances the legitimacy of predetermined decisions, ultimately increasing their effectiveness in informing policy processes." But this is not true if the power imbalances are not considered during this step.

This suggests that 'knowledge' is co-produced first, and only afterward do we reflect on biases and imbalances. There is also a similar inconsistency between dimensions 3.1 and 3.2 (which I will mention later). This highlights how these five dimensions are overlapping and highly interconnected, yet they are presented as a protocol or independent guidelines.

**Response SC29**

We agree with the observations of the reviewer. To answer this comment we need to clarify that our framework is not a protocol, but a conceptual contribution, which can be the basis for a practical protocol, as mentioned in the Response to comment SC19. As a consequence, the dimensions do not have any temporal or consequential order, they are the ingredients that need to be used within a co-creation work and are sometimes intertwined and mutually communicating. For example, in 3.1 we have mentioned that "*Stakeholders' identification is an iterative process, where additional stakeholders are incorporated as the analysis unfolds*."
We acknowledge that this is not clear in the current version of the manuscript and we will include this in section 3:

"The five Each dimensions are the pillars of any transdisciplinary work, and i) include generic elements as well as drought-specific ones and ii) connection between the dimensions showing how they mutually influence each other throughout a co-creation process. "

Also, we changed figure 2 to include a less hierarchical representation of the 5 dimensions, also highlighting the mutual influences of between some of the dimensions are reported in the text (see **Response to comment SC26**).

**SC30** Regarding the lack of specificity in these dimensions, there is a lot of ambiguity or "neither-nor" statements. For instance, the guidelines are neither too well-defined nor too vague, leaving decisions to the "research analyst"—but who is this research analyst, and why are these decisions not co-decided with stakeholders?

 L142 : "Attention should be paid to verifying that these boundaries would not be too restricted to avoid the unintentional overlooking of some stakeholders, leading

to the omission of relevant individuals associated with the phenomenon (Clarkson, 1995). Conversely, the boundaries cannot be too blurred. It is often impractical to include every stakeholder, requiring the establishment of well-founded criteria by the research analyst to determine a cutoff point (Clarke and Clegg, 1998)."

**Response SC30**

As well noted by the reviewer, the decision should be taken by researchers and stakeholders together. We have changed that in the text:

*"It is often impractical to include every stakeholder, requiring the establishment of well-founded criteria by the researchers and stakeholders to determine a cutoff point (Clarke and Clegg, 1998). "*

**SC31** Additionally, the content in the first dimension on mapping stakeholders, particularly about setting boundaries for the study, is made redundant in the second dimension, Section 3.2, from L158 onwards, which also discusses the different types of boundaries.

**Response SC31**

In Section 3.2, we refer to other types of boundaries, which are related to the scope of the co-creation process, therefore they are not redundant with the ones related to the stakeholders.

**SC32** The entire results section should be refocused to remove elements of discussion that are almost philosophical and too vague. The gaps in transdisciplinary approaches are identified far too late and, as I have mentioned earlier, would be better placed earlier in the manuscript, such as in the state of the art or methods section, where they should be presented more clearly rather than in a discussion-like manner.

**Response SC32**

As mentioned in our previous responses, we will revise the introduction to more clearly highlight the gaps in transdisciplinary approaches. We will also address these gaps in the abstract.

Additionally, we will restructure the sections (3.1 to 3.5) that describe the key dimensions, incorporating more examples of applications and ways to integrate each dimension into a transdisciplinary approach for drought impact assessment. We hope these revisions will help make the discussion more focused and concrete, reducing any ambiguity or philosophical overtones.

**SC33** Section 3.4, which addresses conceptualising and implementing the model, should, in my opinion, be the most important section. Yet, it remains extremely

general, overly simplistic, and overly discursive, with elements that are disconnected from reality. Moreover, there are no elements related to actually 'implementing' the model in that section.

**Response SC33**

We understand the concern of the reviewer. We meant to keep the description of this dimension somehow generic to embrace a broader set of models, including for example conceptual models and not just hydrological ones. We understand that this might have left with a sense of superficiality, which was not our ambition. We will improve this section by mentioning a number of examples from different modeling approaches, including elements of model co-selection and co-implementation, although these are a minority of all the reviewed studies.

**SC34** L219 : "From a technical standpoint, co-modelling assumes that, given a suitable interactive environment, non-specialized people can collaboratively produce models that are meaningful to them, fostering valuable discussions and the creation of new knowledge," and this is said to be true "irrespective of the type or purpose of the model" (L218).

But what exactly is a "suitable interactive environment"? Practically, such an environment rarely exists. There are far more limitations to this assumption than conditions that meet it. Without a clear definition of what constitutes a suitable interactive environment, it is an easy way to claim that co-modelling is always effective and possible. In reality, there are never truly 'suitable interactive environments.' Approaches will always face challenges such as power and knowledge imbalances, time and financial constraints, participation limitations, etc.

**Response SC34**

A suitable interactive environment is a space that allows for fruitful interaction between stakeholders enabling them to share and carry on co-creation activities, including co-modeling. Although barriers, errors, and limitations are unavoidable, like in any real-world situations, we provide an overview of the conditions that need to be pursued while setting up a co-creation process. At the same time, we also discuss the limitations mentioned by the reviewer in the discussion to provide an exhaustive account of co-creation experiences from the literature.

**SC35** L230: ok, but then it's not co-modelling anymore.

**Response SC35**

According to a broader definition of co-modeling, also mentioned in the reference therein (Melsen et al., 2018), co-modeling can have a range of depths in the way stakeholders contribute in developing, selecting or setting up the model.

**SC36** L224 : "In transdisciplinary settings, the concept of 'constructing models' can take on diverse interpretations." It can, but the whole point of Table 1 is to provide clear definitions. Therefore, this section should be crystal clear about the guidelines according to the level of participation. Instead, it moves back and forth, without clearly distinguishing between participatory modelling, knowledge co-creation, co-modelling, and different degrees of participation—just floating in the text without a clear thread. It is very difficult to follow, and the purpose becomes unclear.

**Response SC36**

"Constructing models" indeed has several facets, and our aim is not to pick only one of these. Instead, we review them, providing an overview of the major interpretation of the concept, which provides researchers with a wider range of options that can best fit their collaborative space.

**SC37** L240 : what is the base of such statement? Framed as such by what and who? "science and knowledge production processes' is so broad. Do you mean 'co-production'?

**Response SC37**

We will rephrased the first part of the paragraph as follows, to clarify what is meant with knowledge production and why it matters to our framework:

*"The process of developing environmental knowledge is often framed as neutral, objective, and unbiased, particularly in fields that prioritize quantitative methods. However, scholars in science and technology studies (STS) argue that environmental knowledge is also shaped by power dynamics, which influence which forms of knowledge and expertise are recognized as scientifically valid and actionable (Budds, 2009; Goldman et al., 2019; King and Tadaki, 2018; Mukherjee, 2022; Turner, 2011; Zwarteveen et al., 2017). What counts as 'scientific' knowledge is particularly relevant in the context of co-creation for drought assessments, as this approach aims to foster more inclusive and equitable knowledge and, ultimately, more just water management practices (Basco-Carrera et al., 2017; Falconi and Palmer, 2017). Achieving these goals requires reflecting on knowledge production itself - i.e. recognizing biases in what is deemed legitimate knowledge and acknowledging that co-modelling alone does not inherently eliminate existing power dynamics. By explicitly addressing these factors through the lens of STS, the co-creation process is better placed to build a shared commitment to reducing drought risk and establish common goals and strategies."*

**SC38** L240-251: does not add much to this section. It is very general and borad introductive litterature.

**Response SC38**

The relevance of including STS perspectives and the concept of knowledge production in our framework will be discussed in the new "Methods" section. However, we agree that the immediate relevance of this discussion to this section was not clearly outlined. We will thus revise the text to highlight the relevance for "Accounting for knowledge biases", and shorten the section. See revised paragraph above.

**SC39** L253: what is 'elite" stakeholders?

**Response SC39**

We will replace "elite stakeholders" with "powerful stakeholders" to avoid confusion.

**SC40** L255 to L262 identifies quite well some of the major limitations of implementing a transdisciplinary approach, most of which are linked to those who conduct them—referred to as 'elite stakeholders'—and how these limitations often benefit them. However, the proposed solutions to address these limitations from L262 to L266 are overly simplistic. Based on the logic developed earlier, why would these 'elite stakeholders' analyse power relations that ultimately benefit them? Additionally, who would be responsible for the second solution mentioned in L264? If the process is iterative, how would agreement be reached on definitions and scenarios, and how would it prevent one group of stakeholders from favouring its vision or biases over another? In fact, what is more likely to happen is an exacerbation of biases when the same processes are repeated indefinitely. This is why the proposed solutions to counter these limitations are overly simplistic and inconsistent with the rest of the section.

**Response SC40**

We appreciate the reviewer's feedback, and we understand the concerns about the simplicity of these suggestions in relation to the complexities discussed in the text.

In response, we would like to clarify that while powerful stakeholders may indeed benefit from certain decisions or power dynamics in the short term, they could face long-term consequences as these dynamics evolve. For example, the benefits they gain in the immediate context might lead to unsustainable practices in the long run. For example, Gwapedza et al., 2024 show that, if these stakeholders are made aware of these potential long-term impacts, their interests may shift over time, and they may remain interested to be engaged in the co-creation process.

Additionally, regarding the solutions mentioned, we acknowledge the challenge of avoiding the exacerbation of biases in iterative processes and ensuring balanced decision-making. We will expand on this point in the revised manuscript, further clarifying how iterative processes can be designed to be more inclusive and selfcorrecting, with periodic reflection and feedback loops that involve all stakeholders. While the risk of bias or exacerbation of existing power imbalances exists, a well-structured iterative process, with transparent criteria and facilitated dialogue, can reduce these risks over time.

References

Gwapedza, D., Barreteau, O., Mantel, S., Paxton, B., Bonte, B., Tholanah, R., ... & Tanner, J. (2024). Engaging stakeholders to address a complex water resource management issue in the Western Cape, South Africa. Journal of Hydrology, 131522.

**SC41** L271 If, 'positivist hydrologists' is really the targeted audience, I would really suggest a more simple and straightforward structure.

**Response SC41**

Positivist hydrologists, as well as socio-hydrologists are the targeted audience of our article. We are confident that by incorporating the changes suggested in these responses, we can simplify the structure of our work. This will make it more accessible and easier to understand for our intended audience.

**SC42** L272: "hydrologists and others"..so everyone? Also, if the introduction focuses on how drought studies and approaches are highly siloed within the hydrological discipline, it seems contradictory to target the audience by discipline rather than by the reader's purpose..

**Response SC42**

With our article we want to provide hydrologists and socio-hydrologists with a conceptual framework showing the ingredients of a co-creation process for drought impact assessment, highlighting the key elements and limitations of such a complex approach.

**SC43** Table 2, with the core aspects, should be in the results section. Why is this part of the discussion if these are (albeit still quite vague) guidelines or suggestions for implementing each of the dimensions? It is essentially a summary table of Sections 3.1 to 3.5. I suggest adding an extra column on the right for the number of each dimension. Also, as I have mentioned extensively earlier, I don't believe the dimensions and core aspects are independent; they are closely interconnected, with bullet points from one dimension relating to others (e.g., bullet points from Dimensions 1 and 5 complement each other). In my view, it defeats the purpose of co-modelling if, through this framework, we apply Dimensions 1 to 4 to produce something and only afterward, as a final step, reflect on the inherent biases of the tangible outcome of the 'co-production' process.

**Response SC43**

We placed the table in the discussion to give some hints on a possible application of our framework into the development of an operative protocol. The simple protocol outlined in the table is not strictly a result of our literature review, but rather a potential implication in future research, possibly accompanied by a practical case study to showcase its operational usefulness.

As for the interdependence of the dimension, we have addressed that in the **Response to comment SC29** and we will carefully highlight that in the text in section 3.

**SC44** L280: what is the basis of such a statement?

**Response SC44**

With this sentence we intend to say that our framework can support the implementations of co-creation studies, which connects to the research gaps identified in the introduction. In the following lines we give an example of how co-creation studies can better capture local and indigenous knowledge, which in turn forster adaptation to drought.

**SC45** L281: Indigenous knowledge is parachuted here. While it is a valuable suggestion, it should be introduced much earlier in the manuscript, specifically when discussing stakeholder selection.

**Response SC45**

We agree with the reviewer. For this reason we will add a dedicated paragraph in Section 3.1 to highlight the importance of including indigenous knowledge :

*"In addressing water resource challenges, particularly drought, the failure to include marginalized groups—especially indigenous populations and low-income communities—can significantly hinder effective decision-making. Drought conditions often exacerbate existing inequalities, making it even more crucial to incorporate diverse perspectives and traditional knowledge into resource management strategies. Despite claims of inclusive frameworks, available methods frequently fall short of genuinely integrating stakeholder input, leading to decisions that overlook local conditions and perpetuate power imbalances. A systematic approach to engaging all water stakeholders, including those with Indigenous knowledge, is essential for developing more equitable and effective responses to drought and ensuring that resource management is truly reflective of community needs and experiences (Hargrove and Heyman, 2020)."*

**SC46** L284: I do agree, as this is the issue with any conceptual framework that has not been groundtruthed. This framework itself has not been co-constructed, so it is unrealistic to expect it to applied without major challenges.

**Response SC46**

The framework presented in our work has not been co-constructed because it was developed through a review of existing scientific literature on co-creation. The intent of the framework is to provide guidance for hydrologists and socio-hydrologists in planning and conducting transdisciplinary studies for drought impact assessment. While it has not undergone direct co-creation with stakeholders, it is designed to be adaptable and serve as a starting point for future, more context-specific co-creation protocols.

**SC47** L285: "Suboptimal decisions and compromises" are an understatement of what comes next. The limitations addressed impact a significant portion of the entire process. These limitations should have been identified as the initial research gaps on which this framework is built or aims to address. What is the point of constructing a framework based solely on studies from very general fields, which are completely disconnected from drought studies or drought in socio-hydrology? This makes the framework conceptual and biased by the authors, but unable to address the real-world challenges. The whole point of a framework for transdisciplinary approaches should be to deal with these limitations and build upon them.

**Response SC47**

We understand the point raised by the reviewer. The sentence, as it is written, does not properly convey our intended message. We will rephrase it as:

*"Nevertheless, operationalizing this framework into a series of actionable steps and applying it to real case studies may present additional challenges due to the specificities of each context. Therefore, operationalization will require finding compromises to address context-specific constraints and limitations, which are discussed in the next section."*

**SC48** L286: I would remove the term "panacea." I don't think the literature, including on transdisciplinarity, suggests that any approach is a cure-all or universally effective solution.

**Response  SC48**

We chose the word "panacea" to convey a clear message to the reader: that transdisciplinarity has its shortcomings and should not be viewed as a universally effective solution. However, we recognize that the term can be too strong and potentially misleading, as it does not have a direct counterpart in the literature. Therefore, **we will rephrase the section title to  "Limitations and shortcomings of co-creation in drought impact assessment".**

**SC49**

L290: "Particularly in drought impact assessment research"—but where are these specific challenges? What is described is general and not specific to drought.

**Response SC49**

We will improve Sect. 4.2 to provide more drought-specific limitations and challenges.

**SC50** L291: "Including all stakeholders impacted by drought in the co-creation process is of paramount importance." Please have a look at some of the preprints in this special issue, which present case studies of droughts affecting entire regions like California, Chile, Northeast Brazil, and the European drought of 2022. This statement is applicable only for very small areas.

**Response SC50**

We agree with the reviewer that there are some cases in which it might not be possible to involve all the stakeholders impacted by a specific drought event. Nevertheless, in our sentence, we were referring to all the stakeholders inside a given area of interest or community, and not necessarily all those impacted by a certain event. More specifically, we referred to what we introduced in Sect. 3.2 as Impacted Units or Groups, i.e. " the specific entities or populations that will be affected by the study. They can be individuals, households, communities, organisations, or ecosystems."

For this reason, to better clarify, we will rephrased L291 as it follows:

*"Including all relevant stakeholders or their representatives from a given "impacted group" — whether individuals, households, communities, organizations, or ecosystems (Sect. 3.2) — in the co-creation process is crucial. However, in cases where impacts are widespread over large spatial areas, the implementation of a fully representative co-creation process may be challenging due to the large number of people involved. This represents one of the limitations of applying transdisciplinary approaches. Furthermore, even when the number of participants is manageable, the process may still face obstacles due to a lack of economic resources, time constraints, or limited knowledge of the study area."*

**SC51** L296: For whom is the implementation of this framework intended then? If it is for positivist hydrologists and practitioners, how would they navigate these issues? These represent significant limitations to the framework, yet there are no tangible or realistic suggestions provided to address them.

**Response SC51**

We suggested, as a suitable solution, learning from available case studies that might successfully cover some aspects of our framework when assessing drought impacts. (L306-307).

As a potential solution, we have suggested learning from existing case studies that may successfully address certain aspects of our framework in the context of drought impact assessment (L306-307). In fact, we envision investigating specific case studies as the next step in our research to identify if they would provide potential solutions or lessons learned that could help in overcoming some of the open challenges.

In light of this feedback, we believe it would be beneficial to incorporate some of this ongoing work into the manuscript, offering a more solution-oriented perspective that directly addresses the limitations highlighted.

**SC52** L301: what does 'by this time' refer to??

**Response SC52**

It refers to the moment in which the project has already been funded. To be clearer we will replace the expression "By this time" with "By that point" and rephrase the whole concept as it follows:

*"Transdisciplinary studies require that goals and methodologies be collaboratively developed with stakeholders to ensure relevance, buy-in, and effectiveness., However, stakeholders are often only actively engaged after the research has received funding. By that point, the researchers have usually already defined key aspects, such as study goals and methodologies, which limits the scope for stakeholder input and collaboration."*

**SC53** L306: "Learning from available case studies that might successfully cover some aspects of our framework when assessing drought impact" is what this study should have started with: to build a hypothesis before testing and validating it—steps that are lacking in this study. Also, which case studies are being referred to? Moreover, how does validating results based on similar studies lead to genuine learning? It would only reinforce existing biases. I would think the opposite: learning from studies that completely contradict your framework and findings, building on understanding why to address these caveats.

**Response SC53**

Our intention in referencing case studies was not to validate the framework in a conventional sense, but rather to provide practical examples of how certain aspects of the framework have been applied in real-world contexts when assessing drought impacts. These case studies serve more as illustrative examples to highlight

potential challenges and successes in operationalizing the framework, rather than to "test" it in a strict sense.

**SC54** L308 to 310: not adding much and lacking basis.

**Response SC54**

We will revise this section and remove any sentences that may seem repetitive.

**SC55** L310: yet, there is no mention in the text of the importance of the local context in applying such a framework.

**Response SC55**

While the importance of considering the local context is touched upon in the discussion of each dimension of the framework, we will revise the discussion section to further emphasize this aspect.

**SC56** L314: It would have been very beneficial if this framework had aimed to address that. This framework itself, due to its disconnection from reality and its foundation on a very limited number of studies—without building on identified gaps and difficulties in transdisciplinary research applied to drought impacts or assessment—and drawing from fields chosen by the authors without explicit justification... also ends up prioritising familiarity.

**Response SC56**

We understand the reviewer's concerns. In our previous responses, we provided a more detailed explanation of how our framework addresses identified gaps and clarified the reasons behind our selection of specific bodies of literature. We will emphasize these aspects more effectively in the manuscript by rewriting the introduction and including a methods section.

Regarding the limited number of studies referenced, we want to clarify that we did not conduct a systematic literature review. However, we will add additional references to further enrich the literature supporting our dimensions, incorporating valuable suggestions from the reviewers.

**SC57** The conclusion reiterates elements that have been strongly questioned in this review. Logically, if these points are addressed, the conclusion should be rewritten as well.

**Response SC57**

We will clearly **reshape the conclusion** based on the changes made to improve the text based on the comments of both reviewers.

---

## Author Comment (AC2)

**REVIEWER 2**
* * *
**Comment 1**
Firstly, the introduction focuses on drought and drought impact assessments. While I do believe a framework for drought impact assessment would be valuable, it seems that the framework and manuscript here are **not really focused on drought**. The framework and descriptions in the manuscript are quite general and would apply to any water issue not only to drought. There are no specific aspects that only relate to drought.

**Response 1**

We agree that some specific parts of the framework might appear not drought-specific. The identified key dimensions do include some general aspects of transdisciplinary approaches, which can be relevant and applicable beyond drought impact applications since they were built on existing literature on transdisciplinarity. In fact, the purpose of our framework is to adapt generic transdisciplinary frameworks to the complex subject of drought impact assessment, which requires specific processes to be carefully addressed. As a consequence, some dimensions embrace those key features of any robust transdisciplinary process, including, for example, stakeholder engagement, while others are more drought-specific dimensions, for example, "building a shared knowledge on drought", which reflects the complex and context-specific nature of drought compared to other natural hazards.

However, we do believe that an effort to adapt an even more generic dimension to the specific subject of drought can indeed strengthen the relevance of our framework and increase its usefulness. For this reason, we will **revise the description of the different dimensions, highlighting the specific features connected to drought research**. We will also rename some of the dimensions to explicitly include drought-related features. For example, "Mapping and engaging stakeholders" (Section 3.1) will be renamed as "Set up a collaborative space for drought knowledge co-creation." , "Framing the scope of the co-creation process" (Section 3.2) will become "Framing the drought co-creation process", while "Conceptualising and implementing the model" (Section 3.4) will be renamed as "Co-selecting and co-developing models to understand drought impacts".

Sections 3.1 and 3.2 address two dimensions that encompass key features of transdisciplinary processes. These sections would benefit most from revision to emphasize the specific aspects related to drought research. In Section 3.1 we will add the following text to better reframe this dimension from the perspective of drought impact assessment:

*"In addressing water resource challenges, particularly drought, the failure to include*

*marginalized groups—especially indigenous populations and low-income communities—can significantly hinder effective decision-making. Drought conditions often exacerbate existing inequalities, making it even more crucial to incorporate diverse perspectives and traditional knowledge into resource management strategies. Despite claims of inclusive frameworks, available methods frequently fall short of genuinely integrating stakeholder input, leading to decisions that overlook local conditions and perpetuate power imbalances. A systematic approach to engaging all water stakeholders, including those with Indigenous knowledge, is essential for developing more equitable and effective responses to drought and ensuring that resource management is truly reflective of community needs and experiences (Hargrove and Heyman, 2020)."*

Moreover, we will  start Section 3.2 providing more context about the importance of aligning the research questions with societal knowledge demands when addressing drought:

*"Co-creation processes envision that not only methods, research itself, and interpretation of results, but also research questions are developed in partnership. From a transdisciplinary perspective, a crucial aspect of this step is aligning the research questions with societal knowledge demands (Sarewitz and Pielke, 2007). Thus, framing the scope of the co-modelling process is crucial, as societal problems often lack clear boundaries, involve multiple stakeholders, and are deeply interconnected with other challenges, especially when dealing with complex and multifaceted phenomena, such as droughts. To ensure alignment between research questions and societal knowledge demands, it's crucial to have a representative group of engaged stakeholders (see Sect 3.1). Framing the problem and setting the research agenda to encompass diverse understandings and perspectives may require expanding the team to include additional disciplines or engaging with other stakeholders to find the right mix. From a practical perspective, this would lead to an iterative initial phase in which a first set of stakeholders is identified, the scope of the process is framed, and then potential additional stakeholders can be added, requiring further refinement of the scope."*

References:
Hargrove WL, Heyman JM. A Comprehensive Process for Stakeholder Identification and Engagement in Addressing Wicked Water Resources Problems. Land. 2020; 9(4):119. https://doi.org/10.3390/land9040119
Sarewitz, D., & Pielke, R. A. (2007). The neglected heart of science policy: Reconciling supply of and demand for science. Environmental Science & Policy, 10(1), 5–16. https://doi.org/10.1016/j.envsci.2006.10.001

**Comment 2**

In addition, the framework seems to be focused on modelling. Drought impact assessment can take many forms, but the framework describes procedures for a co-modelling approach. This is related to my second point. The introduction describes the differences between co-creation, co-production and co-modelling in detail and

in a way that makes it seem important to consider the whole co-creation process and not only the co-modelling part. However, the description of the framework focuses mostly on co-modelling. It would be nice to see perhaps some other co-development methods aside from co-modelling. Could the co-creation process include a step where the appropriate methods are decided? Rather than focusing on modelling from the start? Alternatively, the paper could be rewritten so that from the start the focus is more on co-modelling, but doing this may be a missed opportunity for introducing hydrologists to transdisciplinary methods that go beyond co-modelling.

**Response 2**

We use co-modeling in a very broad way to accommodate different kinds of modeling approaches. Here, a model is any simplified representation of reality, including mathematical or computer models, but also conceptual models related to hydrological or socio-hydrological aspects. We acknowledge that this definition was not explicit in our submitted version of the manuscript. **We will reshape the dimension "Conceptualising and implementing the model" to include a deeper review of modeling approaches** for drought impact assessment.

**Comment 3**

The description of the framework seems to be more about describing the problems of not taking a transdisciplinary approach and making the case for doing so, rather than describing the actual steps that should be taken to overcome these hurdles and to implement a transdisciplinary approach. An example is section 3.5: lines 240-262 describe how there may be power imbalances between stakeholders and that this can influence model construction and output and only in the last two sentences, in lines 262-266, there is some mention of what steps would be needed for a transdisciplinary approach. This also applies to the other sections describing the framework. Overall, this means that section 3 reads more like an introduction to the difficulties that come with doing transdisciplinary research rather than a description of a framework and concrete steps and examples for overcoming these difficulties. Section 4 introduces some steps for each part of the framework, but it would be nice to see these steps described in more detail, including some examples of how this could be done. These examples are being hinted at in section 4.2: "To propose suitable solutions to these and many other problems, a promising opportunity consists of learning from available case studies that might successfully cover some aspects of our framework when assessing drought impacts." It would be nice to include a review of these examples and include the learning from that in the framework. Including this would make the framework more useful in the sense that it would provide hydrologists with guidance on which transdisciplinary methodologies to implement and how. In its current form, the framework does not really go beyond a description of the problem.

**Response 3**

We really thank the reviewer for suggesting a constructive improvement of Section 4. **We will draw from existing cases in the literature to review and exemplify more concretely the recommendations** and potential problems of knowledge co-creation for drought research. We will start by drawing from the article suggested by the reviewer in their last comment and expand on similar ones. We believe that this would make our dimension much more understandable and practically useful.

**Comment 4**

One part of the framework is about setting a clear scope. However, I would argue setting a clear scope is part of any modelling study and isn't necessary transdisciplinary. The way one does it could be transdisciplinary, but I feel the description is very broad and not focused on the transdisciplinary aspects of setting the scope. In fact, the examples of time periods that are given here are very hydrological (a drought event, for example) and wouldn't allow for, for example, including policy processes that usually take longer than one drought event, but from a transdisciplinary perspective may be very important to include in the analysis.

**Response 4**

We will add a paragraph at the beginning of section 3.2 "Framing the scope of the co-creation process", to better clarify the transdisciplinary aspects related to the phase of scope setting. Here is a proposal for the text we will add:

*"Co-creation processes envision that not only methods, research itself, and interpretation of results, but also research questions are developed in partnership. From a transdisciplinary perspective, a crucial aspect of this step is aligning the research questions with societal knowledge demands (Sarewitz and Pielke, 2007). Thus, framing the scope of the co-modelling process is crucial, as societal problems often lack clear boundaries, involve multiple stakeholders, and are deeply interconnected with other challenges, especially when dealing with complex and multifaceted phenomena, such as droughts. To ensure alignment between research questions and societal knowledge demands, it's crucial to have a representative group of engaged stakeholders (see Sect 3.1). Framing the problem and setting the research agenda to encompass diverse understandings and perspectives may require expanding the team to include additional disciplines or engaging with other stakeholders to find the right mix. From a practical perspective, this would lead to an iterative initial phase in which a first set of stakeholders is identified, the scope of the process is framed, and then potential additional stakeholders can be added, requiring further refinement of the scope."*

Reference:
Sarewitz, D., & Pielke, R. A. (2007). The neglected heart of science policy: Reconciling supply of and demand for science. Environmental Science & Policy, 10(1), 5–16. https://doi.org/10.1016/j.envsci.2006.10.001

Regarding the specific aspect of the time, we agree that indeed **two different time dimensions might be considered,** one is the time horizon of the modeling, and the other one is the time horizon of the transdisciplinary process. **We will rephrase the text to take into account this further aspect.**

**Comment 5**

Finally, section 2 is very short and does not contribute that much to the manuscript. I wonder why only the state in the art for socio-hydrology is described? Why not include sections on the state of the art in transdisciplinary research in sustainability science, integrated water resources management, socio-hydrology, science and technology studies, and political ecology? In addition, I feel section 2 is missing some socio-hydrological studies on participatory approaches. They may not be on drought, but there are several studies on co-creation or at least co-modelling within the socio-hydrological literature. Some of them may be limited in how much they include stakeholders, but I think they should be included in the discussion of the state of the art in socio-hydrology. Some examples include:

https://doi.org/10.1016/j.jhydrol.2024.131522
https://doi.org/10.1016/j.envsci.2023.03.012
https://doi.org/10.1016/j.jhydrol.2024.131248
https://doi.org/10.3390/hydrology9030049
https://doi.org/10.5194/hess-26-5103-2022
https://doi.org/10.1016/j.apgeog.2015.05.008
https://doi.org/10.1016/j.envsci.2015.09.009

**Response 5**

We thank the reviewer for suggesting additional literature. We will carefully review it and include it in the manuscript.

First of all, we would like to clarify that the reason for having a specific state-of-the-art session only for socio-hydrology, lies in the fact that **this paper approaches drought from a socio-hydrological perspective**, offering a framework tailored for transdisciplinary studies and projects that view drought as a result of feedback between water systems and human activities. As a natural consequence, **socio-hydrology is one of the seven bodies of literature we are investigating, and it has been the first from which we began our methodological exploration.** Transdisciplinary approaches in socio-hydrology mainly address flood hazard. Nevertheless, as we intended to do with all the other bodies of literature, we tried to understand if some key features might be transferred to drought.

Clarified that, according to the comments received also by the other reviewer, we have decided to remove Section 2 and move its content into the Introduction, to further contextualize the research gap within the target research field of socio-hydrology.

---

## Editor Decision (ED1)

Thanks for revising the manuscript. The two reviewers have evaluated your revisions and although they both agree that the manuscript has improved significantly, they also still have doubts about the methods, their robustness, their purpose, the intended audience, and the presentation of the results. They specifically question how your analysis is specific for drought.

One of the reviewers suggested a third reviewer, but I do not want to delay the process further, therefore I have done a review of the manuscript myself. I agree with the points made by the reviewers, especially about the paper not being specific to drought. For example, in Sect. 3.2 you did not include any reference to drought literature, when you discuss how that aspect of the framework is relevant for droughts (l.231-233).

In the introduction you state that: "Drought occurrences and impacts are generally considered hydrological extreme phenomena and, thus, are conceptualised and modelled with a hydrological process approach" (l. 36-38) I don't agree with this statement. Most drought impact studies focus more on the social, economic, health-related, or political consequences of drought. And I feel that you are mixing two different things. One: that disasters are not natural and social aspects need to be included in drought impact analysis, and two: that more transdiciplinarity and co-creation is needed. I think the former is well recognised in drought impact research and there are a lot of examples. The latter I feel is done, but not mentioned as such. The literature that you mention in this paper mostly discusses the former and not the latter (for example lines 137-146), which is not in line with the main aim of your paper.

These different angles also become clear in Sect. 4, where you write: "This framework directly addresses three critical aspects, which are currently overlooked in hydrological approaches to drought impact assessment: the underrepresentation of socio-political factors, the exclusion of local knowledge, and the insufficient attention given to the politics of knowledge production." (l.401-403). Here again you assume that drought impact assessment is always based on hydrological approaches. This is not the case. Therefore, the first two points you made here are not valid. "Traditional drought assessments often fail to consider the socio-political contexts that influence vulnerability and impacts." (l.404) and "Moreover, many drought impact assessment studies overlook the knowledge and experiences of local communities, leading to assessments that are disconnected from the perception and expectations of those impacted." (l.411-412) There is no justification of these statement or references to drought literature. The same holds for the conclusions.

In the introduction and methods you state that: "Given the limited literature on transdisciplinary approaches specifically focused on drought …" (l. 97-98) and "Due to the limited literature available in the socio-hydrology field, specifically related to drought, …" (l.118-119) I don't agree that there is not enough literature on socio-hydrological aspects of drought and its impacts. Maybe you need to look beyond Global North publications or go back in time. Key drought researchers like Wilhite, already discussed how drought is interwoven with social processes in the 1970s and 1980s, for example Wilhite & Glantz (1985). Scholars in Africa focus on societal drought impacts and include aspects of transdisciplinary research by design, by building their work on interviews and surveys and by engaging with local experts. For example, Mpandeli et al. (2015) included personal communication of experts.

What keywords did you use for the literature search? Only starting from a few papers and then using snowballing is strongly limiting your reach. There are many drought impact / vulnerability / risk studies that have been developed (partly) based on stakeholder input, but these do not often mention the words co-creation or transdisciplinarity. You need to include also a few key

drought papers and also snowball from there and see if you can add more specific drought-related examples. I also agree with one of the reviewers that maybe your paper should in hindsight be classified as perspective paper, rather than review paper. I will suggest this change to the editorial board, but also want to hear your opinion.

I think you can still make the argument that drought impact research needs more co-creation and transdisciplinarity, but you cannot do this without reviewing the drought impact literature itself. I would be happy to publish a paper in which you first review the drought impact literature (or at minimum, summarise the excellent drought impact reviews that already exist) and then discuss what needs to be changed (based on the other research fields that you have studied and examples from drought studies that have implemented this or examples from drought studies showing how it can be implemented). For example, in Sect. 3.1, you would need to discuss which stakeholders would be specifically important for drought impact studies (beyond the generic category of "marginalised groups", l.191-198), based on drought literature.

Also, in Sect. 3.3 you argue that "there is no single, universally accepted definition of drought (Krueger and Alba, 2022)" (l.273-274). This is actually precisely related to the fact that drought often is defined from the impacts instead of from the hazard. Because the impacts of drought are so diverse, also the hazards that lead to these impacts are diverse and therefore there is no universal definition. This has already been argued by Wilhite & Glantz (1985), Lloyd-Hughes (2014), AghaKouchak et al. (2021) and many others. This needs to be considered a pro instead of a con. So, in Sect. 3.3 you need to review and discuss the body of literature that investigates and discusses how "drought may be experienced and conceptualised" (l.282).

Finally, in Sect. 5, the challenges are formulated very broadly, but also here you should discuss how this is relevant specifically for drought (by citing drought literature that discusses these challenges).

In summary, I think your paper has value and the drought community can learn from the literature you bring in, but it can only be of value if you build on the drought research that is already out there and make the recommendations specific to the drought field. So, please address the points I mentioned here and the points made by the two reviewers as much as you can in a revised version. I will then evaluate this new version.

**Textual comment:** l.130: SÂcience

**References:**

- AghaKouchak, A., Mirchi, A., Madani, K., Di Baldassarre, G., Nazemi, A., Alborzi, A., ... & Wanders, N. (2021). Anthropogenic drought: Definition, challenges, and opportunities.
- Lloyd-Hughes, B. (2014). The impracticality of a universal drought definition. *Theoretical and applied climatology*, *117*, 607-611.
- Mpandeli, S., Nesamvuni, E., & Maponya, P. (2015). Adapting to the impacts of drought by smallholder farmers in Sekhukhune District in Limpopo Province, South Africa. *Journal of Agricultural Science*, *7*(2), 115.
- Wilhite & Glantz (1985). Understanding the Drought Phenomenon: The Role of Definitions.

---

## Author Response (AR2)

**Reply to reviewer comments - Major Revision 2nd round**

We want to thank the editor and the two reviewers for their helpful comments on our manuscript.

We greatly appreciate that the reviewers acknowledged the significant improvements made since the initial submission. Nevertheless, we have taken into account their remaining concerns regarding the methods, their robustness, their purpose, and the presentation of the results. We believe that many of these concerns derive from the fact that the manuscript has been presented as a Literature Review, despite our original intention to develop it more as a Perspective Paper.

Following the suggestion of one of the reviewers (endorsed by the Editor and approved by the Editorial Board), **we are now submitting the revised manuscript as an Invited Perspective**, as reflected in the new title: *"Invited Perspective: Advancing knowledge co-creation in drought impact studies."*

The paper is built on the perspective that, although many drought impact studies involve stakeholders through a variety of participatory approaches, **the bottleneck in current transdisciplinary drought impact research lies in the breadth and depth of knowledge integration.** Most studies either limit co-creation to specific phases of the research process, such as problem definition, scenario development, or result validation (breadth), or fail to meaningfully incorporate non-academic knowledge within those phases, for instance by using such knowledge only to validate predefined scientific assumptions rather than to shape core research questions, methodologies, or models (depth). This leaves room for improving knowledge co-creation with the implementation of more "mature" transdisciplinary work. Drawing from five diverse bodies of literature on transdisciplinarity, we argue that drought impact studies would benefit from the development of a transdisciplinary framework that enables more integrated, power-sensitive, inclusive, situated, and reflexive approaches.

Based on the change in manuscript type and in response to the comments received, we have made **major revisions to the paper**. Specifically, we have:

- **Reshaped the research gap and the rationale for our perspective**, and reflected these changes in the Abstract and the Introduction, including drought-specific literature to support.
- In light of the manuscript's reclassification, we have restructured **the section** previously titled **"Methodology"**. It **is now presented as "*An interdisciplinary perspective on knowledge co-creation"*, focusing on conceptual exploration** rather than methodological procedures. This section offers background on the five selected bodies of literature and discusses how each can enrich transdisciplinary approaches in drought impact studies.
- **Enriched and refined the description of several dimensions** by focusing more closely on the specificities of drought impact studies and by addressing specific comments received during the review process. Table 1, which links each study to the key dimension it informs and to its corresponding body of literature, has been updated to incorporate these new references. In light of the revised format of the manuscript as a Perspective, the table has been moved to the Appendices.
- **Reorganized the Discussion and Conclusion sections** to enhance clarity and coherence. Section 4 addresses the limitations of current transdisciplinary approaches in

drought impact studies, while Section 5 offers insights into how the key dimensions presented in this paper can support the development of more structured and holistic transdisciplinary practices. This includes a **revised version of Table 2 (now Table 1)**, which synthesizes our perspective by highlighting the breadth (key process dimensions) and depth (practical actions for the meaningful inclusion of non-academic knowledge within each dimension) aspects.

Moreover, we revised the entire manuscript to **place less emphasis on introducing a framework** and instead present the dimensions more simply as key elements that, we argue, can support more integrated, power-sensitive, inclusive, situated, and reflexive approaches to drought impact studies. We have also relaxed the **focus of our perspective on drought impact studies, and not only assessment studies**, to include all those studies which work on understanding or investigating these impacts, but without necessarily performing an assessment.

We have addressed the editor and reviewers' feedback. A detailed, point-by-point response to each of their concerns is provided below. We have also **attached the revised manuscript with changes tracked**.

**Line numbers referred to in our responses below are from the revised manuscript (without changes tracked).**

**EDITOR**
* * *
**Comment 1**

Thanks for revising the manuscript. The two reviewers have evaluated your revisions and although they both agree that the manuscript has improved significantly, they also still have doubts about the methods, their robustness, their purpose, the intended audience, and the presentation of the results. They specifically question how your analysis is specific for drought.

One of the reviewers suggested a third reviewer, but I do not want to delay the process further, therefore I have done a review of the manuscript myself. I agree with the points made by the reviewers, especially about the paper not being specific to drought. For example, in Sect. 3.2 you did not include any reference to drought literature, when you discuss how that aspect of the framework is relevant for droughts (l.231-233).

**Response 1**

We sincerely thank the Editor for taking the time to personally review our manuscript.

Regarding the concern that the paper is not sufficiently specific to drought, we have made further efforts to enhance its focus. In particular, we enriched and refined the description of several dimensions by more closely highlighting the specificities of drought impact studies. We added and discussed drought-specific references in all five dimensions (Sections 3.1-3.5), especially to support statements that emphasize drought-related aspects or point out existing gaps in the drought literature concerning those dimensions. These revisions also include the lines in Section 3.2 referenced by the Editor in this comment, where we included references from Rossi et al. (2023) and Hagenlocher et al. (2023) - see ***lines 243-246*** of the revised version.

However, we would like to clarify that drought-specific literature was not made central in each dimension by design, as the objective of the manuscript is not to deliver a literature review on drought per se, but rather to draw from other bodies of literature to support transdisciplinary thinking and enrich the understanding of drought impacts.

**Comment 2**

In the introduction you state that: "Drought occurrences and impacts are generally considered hydrological extreme phenomena and, thus, are conceptualised and modelled with a hydrological process approach" (l. 36-38) I don't agree with this statement. Most drought impact studies focus more on the social, economic, health-related, or political consequences of drought. And I feel that you are mixing two different things. One: that disasters are not natural and social aspects need to be included in drought impact analysis, and two: that more transdiciplinarity and co-creation is needed. I think the former is well recognised in drought impact research and there are a lot of examples. The latter I feel is done, but not mentioned as such. The literature that you mention in this paper mostly discusses the former and not the latter (for example lines 137-146), which is not in line with the main aim of your paper.

**Response 2**

We thank the Editor for this constructive comment. We agree that the research gap was not clear and may have been misleadingly stated that social aspects were not sufficiently considered, or that transdisciplinary studies were absent in the field. We have now revisited the rationale of our perspective to better reflect our original objective and to more clearly highlight the initial research gap that guided our investigation:

*Although many studies involve stakeholders with a variety of participatory approaches, the limitation of current transdisciplinary approaches in drought research lies in the depth and breadth of knowledge integration. Most studies either limit co-creation to specific phases of the research process, such as problem definition, scenario development, or result validation (breadth), or fail to meaningfully incorporate non-academic knowledge within those phases, for instance by using such knowledge only to validate predefined scientific assumptions rather than to shape core research questions, methodologies, or models (depth). This leaves room for improving knowledge co-creation with the implementation of more "mature" transdisciplinary work, building on a full engagement and integration of different stakeholders (and knowledge holders) in all the phases of the research process. Also, and very important, is the attention to power dynamics through the process, which is largely analysed within social sciences, and which can substantially advance socio-hydrological modeling within transdisciplinary research.* (**lines 84-93**)

**Comment 3**

These different angles also become clear in Sect. 4, where you write: "This framework directly addresses three critical aspects, which are currently overlooked in hydrological approaches to drought impact assessment: the underrepresentation of socio-political factors, the exclusion of local knowledge, and the insufficient attention given to the politics of knowledge production. " (l.401-403). Here again you assume that drought impact assessment is always based on hydrological approaches. This is not the case. Therefore, the first two points you made here are not valid. "Traditional drought assessments often fail to consider the socio-political contexts that influence vulnerability and impacts." (l.404) and "Moreover, many drought impact assessment studies overlook the knowledge and experiences of local communities, leading to assessments that are disconnected from the perception and expectations of those impacted." (l.411-412) There is no justification of these statement or references to drought literature. The same holds for the conclusions.

**Response 3**

In response to this concern, we have revisited the rationale of our perspective. The main gap we aim to address is that most available studies either limit co-creation to specific phases of the research process or fail to meaningfully integrate non-academic knowledge within those phases, thereby limiting the overall breadth and depth of these studies. The partial inclusion of local knowledge is particularly relevant in the socio-hydrological literature dealing with co-modeling and the implications of decision-making on model components and assumptions.

Moreover, we focused on the insufficient attention given to the politics of knowledge production. This aspect is emphasized in other bodies of literature, such as hydrosociology, which are often overlooked by socio-hydrologists due to their grounding in different scientific traditions. This further supports our decision to engage with diverse disciplinary perspectives.

In line with this shift in focus, the section discussing the three main gaps (formerly Section 4) has now been completely rewritten to better reflect this updated perspective and to avoid overgeneralizations regarding hydrological approaches to drought assessment.

**Comment 4**

In the introduction and methods you state that: "Given the limited literature on transdisciplinary approaches specifically focused on drought ..." (l. 97-98) and "Due to the limited literature available in the socio-hydrology field, specifically related to drought, ..." (l.118-119) I don't agree that there is not enough literature on socio-hydrological aspects of drought and its impacts. Maybe you need to look beyond Global North publications or go back in time. Key drought researchers like Wilhite, already discussed how drought is interwoven with social processes in the 1970s and 1980s, for example Wilhite & Glantz (1985). Scholars in Africa focus on societal drought impacts and include aspects of transdisciplinary research by design, by building their work on interviews and surveys and by engaging with local experts. For example, Mpandeli et al. (2015) included personal communication of experts.

**Response 3**

We agree that the sentence may have been phrased in an unclear way. Our intention was not to say that there is limited literature on socio-hydrological aspects of drought, but rather that the limitation lies in the rarity of truly transdisciplinary approaches to the investigation of these aspects, which is precisely the motivation behind our perspective. We have now rephrased the introduction, focusing less on the idea of "limited literature" and more on the lack of a deep and comprehensive approach to co-creation of knowledge, integration across disciplines, and the need for greater self-reflection on the process of knowledge production, including attention to power imbalances.

Mpandeli et al. (2015), for example, provide an important early attempt to include non-academic perspectives in drought impact assessments. However, their study still builds on a top-down quantification of drought using hydroclimatic thresholds. We have now included this reference in Section 3.3 (***lines 264-265***) to support our point that engaging in a transdisciplinary research process involves more than including personal communication from local experts—it requires a more structured and integrated approach. This justifies our decision to explore other domains with a longer tradition in knowledge co-creation to learn from them.

Regarding the work of Wilhite & Glantz (1985), we cited it in the text at the beginning of the Introduction (***lines 31-33***), to better explain that socioeconomic and environmental impacts of drought have long been studied, and this does not represent a current gap. However, the actual gap lies in the fact that these impacts and their socio-ecological consequences have traditionally been evaluated and understood primarily as outcomes of a drought hazard conceptualized as a hydrological process. Adopting fully transdisciplinary approaches would allow for a more comprehensive understanding of drought hazard and impacts, including how they are perceived and experienced.

**Comment 5**

What keywords did you use for the literature search? Only starting from a few papers and then using snowballing is strongly limiting your reach. There are many drought impact / vulnerability / risk studies that have been developed (partly) based on stakeholder input, but these do not

often mention the words co-creation or transdisciplinarity. You need to include also a few key drought papers and also snowball from there and see if you can add more specific drought-related examples. I also agree with one of the reviewers that maybe your paper should in hindsight be classified as perspective paper, rather than review paper. I will suggest this change to the editorial board, but also want to hear your opinion.

**Response 5**

Following your suggestion and that of another reviewer, we have reclassified the paper as a perspective rather than a review article and have accordingly removed the detailed explanation of our previous review methodology from the main text.

However, to clarify our original approach, we initially focused our literature search on drought-specific studies that explicitly referenced transdisciplinary or co-creative approaches. We found that relatively few of these works implemented fully transdisciplinary processes, particularly concerning iterative feedback loops and considerations of power dynamics. This observation led us to broaden our scope and explore literature from other domains that more thoroughly addressed these aspects.

In response to your feedback, we have now added a section in the Introduction that highlights current transdisciplinary approaches in drought research. This provides a clearer rationale for the shift in focus and strengthens the context for our perspective.

**Comment 6**

I think you can still make the argument that drought impact research needs more co-creation and transdisciplinarity, but you cannot do this without reviewing the drought impact literature itself. I would be happy to publish a paper in which you first review the drought impact literature (or at minimum, summarise the excellent drought impact reviews that already exist) and then discuss what needs to be changed (based on the other research fields that you have studied and examples from drought studies that have implemented this or examples from drought studies showing how it can be implemented). For example, in Sect. 3.1, you would need to discuss which stakeholders would be specifically important for drought impact studies (beyond the generic category of "marginalised groups" , l.191-198), based on drought literature.

**Response 6**

We have now further improved the overview of available transdisciplinary approaches in drought impact studies in the introduction (*lines 61–83*), adding five additional works, summarising what already exists, and highlighting their contribution to the field.

We have also made a further effort to include more drought-specific elements in each dimension. Specifically, in Sect. 3.1, we have now included a discussion about which types of stakeholders are currently involved in transdisciplinary drought studies, how they are involved, and the limitations of their involvement (*lines 163–175*), including nine drought-related citations.

**Comment 7**

Also, in Sect. 3.3 you argue that "there is no single, universally accepted definition of drought (Krueger and Alba, 2022)" (l.273-274). This is actually precisely related to the fact that drought often is defined from the impacts instead of from the hazard. Because the impacts of drought

are so diverse, also the hazards that lead to these impacts are diverse and therefore there is no universal definition. This has already been argued by Wilhite & Glantz (1985), Lloyd-Hughes (2014), AghaKouchak et al. (2021) and many others. This needs to be considered a pro instead of a con. So, in Sect. 3.3 you need to review and discuss the body of literature that investigates and discusses how "drought may be experienced and conceptualised" (l.282).

**Response 7**

We agree that the lack of a unique, universally accepted definition of drought is not a disadvantage. Indeed, it was not our aim to assess whether this is positive or negative. Rather, we mentioned this aspect to highlight that, since no universal definition exists, it becomes essential in a co-creation process to build a shared understanding of drought and its impacts among the relevant knowledge holders. This shared understanding can also involve the recognition of multiple perspectives, derived from different conceptualisations and knowledge systems.

We have now clarified this point in Sect. 3.3 (**lines 261-264**): that *"It is well recognised that there is no single, universally accepted definition of drought (Krueger and Alba, 2022). While hydrological sciences are often rooted in positivist paradigms, there is growing recognition that drought is a complex phenomenon arising from the interplay between biophysical and socio-economic factors (AghaKouchak et al., 2021; Wilhite and Glantz, 1985)."*

**Comment 8**

Finally, in Sect. 5, the challenges are formulated very broadly, but also here you should discuss how this is relevant specifically for drought (by citing drought literature that discusses these challenges).

**Response 8**

Given the change in the manuscript type, we have fully revised the final part of the manuscript, which no longer includes discussion and conclusion sections. Additionally, the three challenges mentioned previously are no longer discussed in the current version of the manuscript.

**Reviewer 1**
* * *
**Comment 1**

Despite the authors' clear and striking efforts to address the comments, I still have strong doubts about the methods, their robustness, their purpose, the intended targeted audience, and the presentation of the results.

There are many aspects I fail to understand. I still do not see how the central five dimensions were deduced or how they are more innovative than common knowledge. I also struggle to grasp the utility of this framework. It is unclear to me what kind of knowledge it aims to produce, and I find terms such as 'knowledge' and 'modeling' too open-ended, and the definition of 'risk impact assessment' too broad

This broadness clashes with the framework's highly ambitious goals. If the terms are so general, it becomes difficult to see how they can effectively address the vast and complex field of risk impact assessment the way the authors define it. Moreover, claiming that this framework can help create 'any knowledge' or support 'any type of modelling' feels overly aspirational, especially when the foundational elements lack specificity and rigour.

Below are some comments regarding the authors' responses, which evoked my previous arguments.

**Response 1**

We thank the Reviewer for acknowledging our efforts. We believe that, by changing the manuscript type from Perspective to Review and introducing substantial revisions, including a more clearly framed explanation of the study's goal, we are now better able to clarify why some parts of our discussion may appear "broad" or not strictly "drought-specific."

Our work aims to improve the comprehensiveness of current transdisciplinary approaches to drought impact assessment, by drawing inspiration from other bodies of literature in which certain aspects—such as self-reflection on the knowledge creation process, the use of iterative processes to ensure the participation of all relevant stakeholders including underrepresented and marginalized groups, and attention to power imbalances—are more explicitly addressed.

These elements, while perhaps more common in other scientific traditions, are still underexplored in many participatory drought-related studies. This justifies our broader perspective and supports the relevance of the proposed framework, even if it may appear ambitious at first glance.

**Comment 2**

Page 1, response 1
"As a result, we identified a first set of recurrent themes among these disciplines, which can be considered as key dimensions to define a cocreation process for drought impact assessment"

There is a dissonance with the table that follows—what recurring themes? How do they lead to the five dimensions, or are they the same? I do not understand whether the authors developed these five pillars *based on lessons and analysis of the papers and their similarities*, or if these five dimensions *are* the similarities the authors observed in these papers. To me, these are two different things.

**Response 2**

We understand that the methodological process we presented might not have been clear enough, and that the choice of the expression "recurring themes" was possibly misleading. To clarify, the "recurring themes" indeed were exactly the five key dimensions we identified. These dimensions were derived from our analysis of shared elements and practices, and insights across the relevant five bodies of literature we investigated.

Now that the paper is presented as a perspective, we have removed the potentially confusing methodological explanation. In any case, the five key dimensions represent essential elements that should be further emphasized and explored in transdisciplinary drought impact studies, particularly to *"support hydrological and socio-hydrological modellers and practitioners in developing more structured, power-sensitive, inclusive, situated, and reflexive co-created drought impact studies."* (***lines 94-95***).

**Comment 3**

Page 2 response 1, 2): How many articles were analysed? It is mentioned, "the individual article contributing to the conceptualisation of each dimension to to their respective body of literature." Is there a typo here, with an "S" missing from "articles"? If it is only one article per dimension and per field (totalling 25 articles), can a solid argument that it is a robust sample size to deduce a dimension, be added?

**Response 3**

We apologize for the confusion, and we confirm that there was indeed a typo in our previous response. The correct phrasing should refer to "articles" in the plural form. To clarify, we have reviewed more than 27 papers to inform the development of the five dimensions, as evidenced by the citations included in the main text of each dimension and the references compiled in the former Table 1 (now moved to the Appendix as Table A1).

Since the paper is now framed as a perspective article, the emphasis lies less on the sample size in a systematic sense and more on the conceptual richness and relevance of the literature selected. Nonetheless, in response to the reviewers' and the editor's feedback and as part of the revision process, we have further expanded our literature base. As of this revised version, a total of 86 papers have been analysed and cited across the dimensions, as documented in the updated table A1.

**Comment 4**

Page 2, Table:

The first column appears to present a result, but I don't believe we are at that stage yet. Here, it seems to imply the existence of these five dimensions upfront and then associate papers that fit

into these dimensions based on their field of study. This approach feels like validating a hypothesis or assumptions the authors already have, rather than elaborating on or deriving the dimensions from the data. While I agree that this structure might bring some clarity to the presentation of the results, it feels more like a band-aid solution to a larger problem. The same problem I highlighted in the previous round of review: What is the framework of analysis? On what basis is it deduced that these five dimensions are the pillars of an important transdisciplinary framework? If the first column is intended to present the rationale behind the dimensions, then these dimensions should emerge as the results. This would mean situating them within the cases, with the corresponding articles linked to the nexus of "field X similarity." Alternatively, if it is assumed that these five dimensions are the starting point (i.e., the first column), I would expect not just citations of the papers that fit into these dimensions, but also the content within those papers that supports the conclusion that each dimension is a pillar of a robust transdisciplinary framework applicable to drought impact assessment.

**Response 4**

Thank you for this important comment. We acknowledge the concern regarding the presentation of the five dimensions and the perceived lack of an explicit analytical framework.

Since the paper has been repositioned as a perspective, it does not claim to present empirical results derived from a systematic review or case-based framework. Instead, it aims to offer a conceptual contribution by proposing five interrelated dimensions that we believe are essential for advancing transdisciplinary approaches to drought impact assessment.

These dimensions were not pre-defined arbitrarily but were shaped through an iterative process of synthesising insights from a broad and interdisciplinary body of literature (now comprising 86 papers). The literature associated with each dimension was selected based on its conceptual contribution and relevance to the respective themes, rather than through a coding or categorisation process typical of empirical content analysis.

In this light, the first column of the table is not meant to present results in the traditional sense, but rather to summarise the proposed conceptual dimensions that structure our perspective. To address the comment and avoid confusion, we have clarified this purpose in the main text and adjusted the language in the table caption accordingly.

**Comment 5**

Page 7, response 4 " Therefore, it would be contradictory to presuppose in advance the specific knowledge we aim to create."

I see your point, but I do not entirely agree. This could be read as a justification for transdisciplinary research to start without a clear sense of what knowledge it aims to generate. Even in transdisciplinary research, there must be an overarching research question or problem statement guiding the process. If knowledge were to "emerge" entirely unpredictably, how would researchers ensure that the findings are useful, actionable, or aligned with the intended goals? Also, most transdisciplinary research is funded based on predefined objectives, expected outcomes, and impact assessments. If knowledge production were entirely open-ended, it would be nearly impossible to justify investments in research. Why, in the first place, would such actors be included in the "knowledge co-creation process" if their contribution to a presupposed topic were not expected? You make an argument in favor of that a few lines below, mentioning that you will specify which steps of modeling might benefit from a transdisciplinary approach. I am

not arguing in favor of assuming exactly what knowledge will be produced, but the way it is phrased suggests that any production of knowledge or connection in the process is entirely open-ended, which is not realistic.

**Response 5**

We agree with the reviewer that a preliminary research question is always necessary to initiate a transdisciplinary process. This initial question may stem from urgent needs, real-world challenges, or concerns about future risks carried on by researchers or other stakeholders, depending on the case. As discussed in the manuscript, the preliminary question can be revisited and refined through the integration of the interests, perspectives, and knowledge of the actors involved in the process.

The research question is essential to start the process of knowledge co-creation. However, this knowledge does not need to exist a priori; rather, it should emerge as an outcome of the transdisciplinary process itself.

**Comment 6**

Page 7 response 4:

"Regarding the concept of co-modeling'
I am really not convinced. I do not see the link with modeling. The argument initially concerns the co-creation of "any knowledge" without presuppositions, but now it shifts to specific steps of modeling. Moreover, the authors use "co-modeling" too broadly—how is that different from "knowledge" in the argument above? To me, this feels off focus. If the claim is that transdisciplinarity could benefit any modeling process, then this argument should be made first. However, this is a separate practical gap from the one addressed in this study. The current approach takes the focus away. I would strongly recommend that the authors focus on one of the following themes: How transdisciplinarity can contribute to modeling, which 'might' have so far remained siloed in hydrology. OR How this review of different literature bodies on transdisciplinarity highlights five common dimensions of a strong co-creation framework applicable to drought risk assessment. At the moment, the paper moves in too many directions.

**Response 6**

We agree with the reviewer that in the previous version, the manuscript's focus may have been unclear, as the five dimensions were presented more as a literature review. The article is now clearly framed as a perspective that aims to explore how to enhance both the breadth and depth of transdisciplinary approaches in drought impact studies. Our objective is to contribute specifically to the advancement of drought impact research, where some form of modeling is always involved.

For this reason, one of the dimensions we explore is modeling, understood in a broad way that encompasses not only quantitative approaches but also more qualitative and interpretive ones. In this broader perspective, modeling is considered a major part of the co-creation of most knowledge related to drought impacts.

**Comment 7**

Page 8, Response 5

"In this paper, the term "drought impact assessment" is used to refer generically to studies and projects that not only evaluate the hazard dimension of drought but also assess its impacts and support the identification and planning of drought management or adaptation measures."

It is a bit contradictory to me when, in SC 15, it is mentioned that governance and policy issues are outside the scope of the study's focus.

**Response 7**

We believe there may have been a misunderstanding. We intended to clarify that the literature you suggested primarily addresses governance and policy issues related to water resources, but does not place a strong emphasis on studies assessing drought impacts. That said, we acknowledge that drought governance and policy can indeed emerge as outcomes of transdisciplinary drought impact studies, when this is among the study's aims. To reflect this, we have also relaxed the focus of our perspective on drought impact studies, and not only assessment studies, to include all those studies which work on understanding or investigating these impacts, but without necessarily performing an assessment.

**Reviewer 2**
* * *
**Comment 1**

The structure and readability of the manuscript has improved significantly. However, I still feel like the framework is quite general and fail to see how it applies specifically to drought. The authors have rephrased some of the dimensions so it now includes "drought", but in describing the framework it is still not clear how these different dimensions are specific for drought and how they should be addressed for drought specifically (as opposed to how they are addressed for another problem). It seems to me the framework could apply to any water problem.

**Response 1**

We thank the reviewer for acknowledging our effort in improving the manuscript. The description of the different dimensions might appear general, since it is mainly based on non-drought-specific literature. The reason for that is that our perspective is based on the analysis of other bodies of literature to learn how to address transdisciplinary drought studies with more breadth and depth.

Clarified that, we have now made a further effort to include more drought-specific elements in each dimension. See e,g,. *lines 163–175, 234-238, 243-246, 264-268, 354-356, 362-363.*

**Comment 2**

In addition, while I acknowledge that examples have been added to the description of the different dimensions and this description has now significantly improved, I still believe that more concrete descriptions of the actions and how to go about implementing the framework is missing. The authors mention that "while it provides a conceptual foundation, it is not a predefined operational protocol or a linear, step-by-step guide. Instead, it highlights essential considerations for protocol development, offering a flexible structure that can guide the creation of tailored protocols." But I feel that without this, the manuscript doesn't make a substantial contribution to the existing literature.

**Response 2**

We thank the reviewer for sharing his valuable perspective. As mentioned, the original aim of the manuscript was not to provide a practical, predefined protocol, but rather to offer high-level guidance for improving transdisciplinary approaches to drought impact studies. We intentionally avoided proposing a fixed set of actions to encourage reflection and offer a flexible perspective that can be adapted to diverse contexts and situations. Indeed, given the need for adaptability across multiple settings, we believe it is appropriate not to include specific descriptions of concrete actions.

Nonetheless, we believe our perspective still provides a valuable contribution to the literature, particularly by reflecting on the current limitations in depth and breadth of current transdisciplinary drought impact studies, especially the lack of attention to power imbalances and self-reflection on the knowledge co-production process.

---

## Editor Decision (ED2)

**De Angeli – revised version**

Thanks for your thorough revision of the manuscript and the clear explanation / justification in the rebuttal. I agree with the points you make in the rebuttal and see the paper now as a more useful contribution to the field. There are a few areas where I see some small but important room for improvement. If you address the following four points, I will be happy to accept the paper. (Line numbers below refer to the revised version of the manuscript without tracked changes.)

**1)** I still think your statements about the field of drought research are not always fully correct and you do not always provide justification for your statements (e.g. l.179-180: "methods frequently fall short of genuinely integrating stakeholder input"), but I also recognise that an outsider perspective can be useful and an perspective paper does not require a full review of the current state-of-the art.

One example where you can easily make a change is L.32-33: "drought hazards primarily modelled as hydrological processes (Mishra and Singh, 2010, 2011)" > please add "meteorological", because many drought (impact) studies even skip hydrology and go directly from the climatic anomaly to the impacts, so this sentence should be rephrased to: "modelled as meteorological and/or hydrological processes"

**2)** The way that you mix up drought (impact) research with modelling is still confusing. In the rebuttal you write that in response to Reviewer 1, comment 6: "one of the dimensions we explore is modeling, understood in a broad way that encompasses not only quantitative approaches but also more qualitative and interpretive ones. In this broader perspective, modeling is considered a major part of the co-creation of most knowledge related to drought impacts." But qualitative approaches is not what the reader would think when you talk about modelling.

In the revised paper you write: "In the field of integrated water resource management knowledge co-creation is often addressed by referring to the concepts of collaborative modelling or co-modelling (Basco-Carrera et al., 2017). This concept involves the collaborative construction of models, which can be physical, conceptual, or computational representations of a system, process, or phenomenon." (l.55-58). Good that you add this definition and I agree that co-modelling can be one aspect that is sometimes used in co-creation, but in the next paragraph you then write:

"… co-modelling of i) drought impacts, ii) water infrastructure planning, and iii) water use under scarcity conditions. The first body of literature includes studies aimed at increasing stakeholders' participation in drought plans with a variety of approaches." (l.61-63). The examples you give in the paragraph after are not all modelling studies, not even with the broad definition you gave before.

I think it is best to leave the modelling focus in Section 3.4 and frame the rest of the paper more broadly. So I suggest that you replace the word "modelling" with knowledge, research, analysis, or something similar, and co-modelling with co-creation (you now seem to use them interchangeably) throughout the manuscript except when you explicitly mean quantitative numerical modelling. For example on l.32, 49, 94, 217, 228, 255-256, 271, 359. The definition and explanation in lines 302-307 are helpful and this applies well to Section 3.4, but the other sections should be more general. In the Introduction you can write that in the perspective you discuss drought impact studies in general first and then zoom in to focus on one approach that is often used, namely modelling.

**3)** In line 115, you list the two last disciplines as "Critical Water Studies, and Science and Technology Studies (STS)", but in line 126 you state that Critical Water Studies is part of Science and Technology Studies (STS) and in line 134 you mention the wider STS. This is confusing (and as a side-note, you don't need to introduce the abbreviation STS twice). What would help is turning the discussion of the dimensions of Critical Water Studies and STS around, so that you can start with the broader STS and then zoom in to Critical Water Studies. If that is difficult, you need to at least change l.134 from "STS provides an insightful self-reflection" to "The wider field of STS provides an insightful self-reflection".

**4)** In line 140, you need to refer back to the previous section by saying that the five dimensions are based on the five research fields highlighted in the previous section. This was already pointed out by Reviewer 1 in Comment 2.

**Textual comments:**

- L.103: "we identified and discussed" > "we identify and discuss"
- L.109-110: "Next, in Sect. 4, we discuss the limitations of transdisciplinary approaches to drought" > "Next, in Sect. 4, we discuss the limitations of transdisciplinary approaches to drought, related to the five key dimensions"
- L.222: "team" > do you mean stakeholders? Or research team? Or a combination? Please specify.
- L.248: "it's crucial" > "it is crucial"

---

## Author Response (AR3)

**Reply to Editor's comments – Minor Revision**

We would like to thank the editor for her minor but very relevant comments on our updated manuscript. We have addressed all of them. A detailed, point-by-point response is provided below. We have also attached the **revised manuscript with changes tracked**.

**Line numbers referred to in our responses below are from the revised manuscript (without changes tracked).**

**Comment 1**

1) I still think your statements about the field of drought research are not always fully correct and you do not always provide justification for your statements (e.g. l.179-180: "methods frequently fall short of genuinely integrating stakeholder input"), but I also recognise that an outsider perspective can be useful and an perspective paper does not require a full review of the current state-of-the art.

One example where you can easily make a change is L.32-33: "drought hazards primarily modelled as hydrological processes (Mishra and Singh, 2010, 2011)" > please add "meteorological", because many drought (impact) studies even skip hydrology and go directly from the climatic anomaly to the impacts, so this sentence should be rephrased to: "modelled as meteorological and/or hydrological processes"

**Response 1**

We agree that our original statement that "*methods frequently fall short of genuinely integrating stakeholder input*" was too subjective and not sufficiently supported by references. We have now slightly rephrased this statement and supported it with relevant literature:

*"Yet, we argue that available methods remain rooted in a "functionalist orientation and a distinct preference for quantification", limiting their ability to meaningfully integrate stakeholder input (Bachmair et al., 2016; Lemos and Morehouse, 2005; Venot et al., 2022, p. 92). As a result, these methods often lead to decisions that neglect local conditions and reinforce existing power imbalances."* L 181-184

Added references:

- Bachmair, S., Stahl, K., Collins, K., Hannaford, J., Acreman, M., Svoboda, M., Knutson, C., Smith, K. H., Wall, N., Fuchs, B., Crossman, N. D., and Overton, I. C.: Drought indicators revisited: The need for a wider consideration of environment and society, WIREs Water, 3, 516–536, https://doi.org/10.1002/wat2.1154, 2016.
- Lemos, M. C. and Morehouse, B. J.: The co-production of science and policy in integrated climate assessments, Glob. Environ. Change, 15, 57–68, https://doi.org/10.1016/j.gloenvcha.2004.09.004, 2005.
- Venot, J.-P., Vos, J., Molle, F., Zwarteveen, M., Veldwisch, G. J., Kuper, M., Mdee, A., Ertsen, M., Boelens, R., Cleaver, F., Lankford, B., Swatuk, L., Linton, J., Harris, L. M., Kemerink-Seyoum, J., Kooy, M., and Schwartz, K.: A bridge over troubled waters, Nat. Sustain., 5, 92, https://doi.org/10.1038/s41893-021-00835-y, 2022.

In addition, following your suggestion, we have revised L.31–33 to read:

*"Socioeconomic and environmental impacts of drought have long been studied (e.g., Wilhite and Glantz, 1985) based on drought hazards primarily assessed as meteorological and/or hydrological processes (Mishra and Singh, 2010, 2011)."*

**Comment 2**

2) The way that you mix up drought (impact) research with modelling is still confusing. In the rebuttal you write that in response to Reviewer 1, comment 6: "one of the dimensions we explore is modeling, understood in a broad way that encompasses not only quantitative approaches but also more qualitative and interpretive ones. In this broader perspective, modeling is considered a major part of the co-creation of most knowledge related to drought impacts. " But qualitative approaches is not what the reader would think when you talk about modelling. In the revised paper you write: "In the field of integrated water resource management knowledge co-creation is often addressed by referring to the concepts of collaborative modelling or co-modelling (Basco-Carrera et al., 2017). This concept involves the collaborative construction of models, which can be physical, conceptual, or computational representations of a system, process, or phenomenon. " (l.55-58). Good that you add this definition and I agree that co-modelling can be one aspect that is sometimes used in co-creation, but in the next paragraph you then write: "… co-modelling of i) drought impacts, ii) water infrastructure planning, and iii) water use under scarcity conditions. The first body of literature includes studies aimed at increasing stakeholders' participation in drought plans with a variety of approaches. " (l.61-63). The examples you give in the paragraph after are not all modelling studies, not even with the broad definition you gave before.
I think it is best to leave the modelling focus in Section 3.4 and frame the rest of the paper more broadly. So I suggest that you replace the word "modelling" with knowledge, research, analysis, or something similar, and co-modelling with co-creation (you now seem to use them interchangeably) throughout the manuscript except when you explicitly mean quantitative numerical modelling. For example on l.32, 49, 94, 217, 228, 255-256, 271, 359. The definition and explanation in lines 302-307 are helpful and this applies well to Section 3.4, but the other sections should be more general. In the Introduction you can write that in the perspective you discuss drought impact studies in general first and then zoom in to focus on one approach that is often used, namely modelling.

**Response 2**

Thank you for this valuable suggestion. We understand the confusion caused by our broad use of the term "modelling" and agree that it may not always align with how readers typically interpret it.

As suggested, we now use broader terms such as knowledge, research, or analysis throughout most of the paper, and reserve the term co-modelling for cases where it clearly refers to modelling, specifically in Section 3.4. We also replaced "co-modelling" with "co-creation" in sections where we refer more generally to participatory or collaborative knowledge processes.

We have made corresponding edits on lines.

- **32** (modelled -> assessed)
- **49** (modelling -> studies)
- **63-64** (co-modelling -> co-creation)
- **94** (modelling -> studies),
- **219-220** (co-modelling -> co-creation)
- **231-232** (co-modelling -> co-creation)
- **260** (co-modelling -> co-creation)

- **275** (co-modelling -> co-creation)
- **363** (co-modelling -> co-creation)

Moreover, in the Introduction, we now explain that the paper first discusses drought impact studies more generally, and later zooms in on modelling as one particular approach often used in co-creation:

*" In this perspective paper, we discuss co-creation in drought impact studies from a broad standpoint and return to co-modelling in Sect. 3.4 to examine its specific role and applications in greater detail, presenting co-modelling as one key dimension of the co-creation process."* (L 60-62)

**Comment 3**

3) In line 115, you list the two last disciplines as "Critical Water Studies, and Science and Technology Studies (STS)" , but in line 126 you state that Critical Water Studies is part of Science and Technology Studies (STS) and in line 134 you mention the wider STS.
This is confusing (and as a side-note, you don't need to introduce the abbreviation STS twice). What would help is turning the discussion of the dimensions of Critical Water Studies and STS around, so that you can start with the broader STS and then zoom in to Critical Water Studies. If that is difficult, you need to at least change l.134 from "STS provides an insightful self-reflection" to "The wider field of STS provides an insightful self-reflection".

**Response 3**

We apologise for this confusion, which was just created by a mistake we made. Indeed, Critical Water Studies are not part of STS. We have just removed the wrong statement at line 126.

**Comment 4**

4) In line 140, you need to refer back to the previous section by saying that the five dimensions are based on the five research fields highlighted in the previous section.
This was already pointed out by Reviewer 1 in Comment 2.

**Response 3**

We have now included a statement at the beginning of Section 3 to create a clearer link between the five bodies of literature of Section 2 and the dimensions discussed in Section 3:

*"We discuss five key dimensions that provide coherent theoretical guidance for advancing transdisciplinary approaches in drought impact studies. These dimensions are derived from the five research fields introduced in Sect. 2 and reflect how insights from these disciplines offer critical perspectives on the role of knowledge integration, power dynamics, and ethical considerations in co-creation approaches for drought impact studies."* (L 140-143)

**Minor Comments**

Textual comments:
- L.103: "we identified and discussed" > "we identify and discuss"

Fixed

- L.109-110: "Next, in Sect. 4, we discuss the limitations of transdisciplinary approaches to drought" > "Next, in Sect. 4, we discuss the limitations of transdisciplinary approaches to drought, related to the five key dimensions"

Fixed

- L.222: "team" > do you mean stakeholders? Or research team? Or a combination? Please specify.

The term stakeholders is used in the paper to address stakeholders in general, including both academic and non-academic. We agree that the term 'team' was not introduced before and could be misleading. Therefore, we rephrased as: "*the co-creation team, including both academic and non-academic stakeholders*".

- L.248: "it's crucial" > "it is crucial"

Fixed